# Interlocking of co-opted developmental gene networks in *Drosophila* and the evolution of pre-adaptive novelty

Sara Molina-Gil [1,2,5], Sol Sotillos [1,5], José Manuel Espinosa-Vázquez[1,3], Isabel Almudi [1,4] & James C.-G. Hombría [1] ✉

The re-use of genes in new organs forms the base of many evolutionary novelties. A well-characterised case is the recruitment of the posterior spiracle gene network to the *Drosophila* male genitalia. Here we find that this network has also been co-opted to the testis mesoderm where is required for sperm liberation, providing an example of sequentially repeated developmental co-options. Associated to this co-option event, an evolutionary expression novelty appeared, the activation of the posterior segment determinant Engrailed to the anterior A8 segment controlled by common testis and spiracle regulatory elements. Enhancer deletion shows that A8 anterior Engrailed activation is not required for spiracle development but only necessary in the testis. Our study presents an example of pre-adaptive developmental novelty: the activation of the Engrailed transcription factor in the anterior compartment of the A8 segment where, despite having no specific function, opens the possibility of this developmental factor acquiring one. We propose that recently co-opted networks become interlocked, so that any change to the network because of its function in one organ, will be mirrored by other organs even if it provides no selective advantage to them.

It has been observed that genes playing particular roles during organ development can be recruited to perform novel functions in other organs. The re-use, or co-option, of developmental genes in new organs, is the base of many evolutionary novelties. An example of this is the highly transparent crystallin proteins that refract light in the eye lens. In all vertebrates, α-*crystallin* evolved from the co-option of a small heat shock protein to the eye; while in birds δ-*crystallin* evolved from the co-option of a different protein, the Arginosuccinase lyase involved in arginine biosynthesis[1].

Although there is abundant research on single-gene co-option, few studies have considered the functional consequences of full-gene network co-option. One of the best-characterised co-option cases is the recruitment of the appendage-forming gene network to form the eye-spots that decorate butterfly wings in several species[2]. Similarly, in the *Drosophila melanogaster* subgroup, the larval respiratory posterior spiracles and the adult male genitalia share the expression of numerous genes due to the recent co-option into the male genital disc primordium of a pre-existing gene network controlling the formation of the external larval respiratory organs[3]. The co-option of this gene network to the male genitalia resulted in the formation of the posterior lobe, a structure present in *D. simulans* and *D. mauritiana*, closely related to *Drosophila melanogaster*, but not in the more distant *D. biarmipes* or *D. ananassae* species[3].

[1]Centro Andaluz de Biología del Desarrollo (CABD), CSIC-JA-UPO Ctra. de Utrera, km1, 41013 Seville, Spain. [2]Present address: Málaga Biomedical Research Institute and Andalusian Centre for Nanomedicine and Biotechnology Platform, Severo Ochoa, 35, 29590 Málaga, Spain. [3]Present address: Department of Food Biotechnology, Instituto de la Grasa. Campus de la Universidad Pablo de Olavide. Ctra. de Utrera, km. 1, 41013 Seville, Spain. [4]Present address: Department of Genetics, Microbiology and Statistics and Institut de Recerca de la Biodiversitat (IRBio), Universitat de Barcelona, Diagonal, 643, 08028 Barcelona, Spain. [5]These authors contributed equally: Sara Molina-Gil, Sol Sotillos. ✉e-mail: jcashom@upo.es

The formation of the posterior spiracles and the posterior lobes have been well-studied in *D. melanogaster*. Posterior spiracle organogenesis is regulated by a gene network activated in the eighth abdominal larval segment (A8) by the Hox protein Abdominal-B (Abd-B) (Fig. 1a, b)[4]. The internal spiracular chamber is formed from A8 anterior compartment cells (A8a) when Abd-B activates in the dorsal ectoderm the transcription of the JAK/STAT signalling pathway ligand Unpaired (Upd) as well as the Empty spiracles (Ems) and the Cut (Ct) transcription factors. The external protruding stigmatophore is formed from both anterior and posterior compartment A8 cells when Abd-B activates the Spalt (Sal) transcription factor (Fig. 1b), which in turn activates *engrailed* (*en*) transcription in a unique A8 pattern[5,6]. These primary factors activate the *RhoGAP Cv-c* and *RhoGEF64C* cytoskeletal regulators, the cell polarity gene *crumbs* (*crb*) and various Cadherins[6]. Abd-B also modulates the expression of *wingless* and the EGF regulator *rhomboid* genes generating A8 specific segmental information distinct from that in more anterior segments[5].

The posterior lobe is a hook-shaped structure used by the male to grasp the female during mating, which may act as a prezygotic reproductive isolation barrier facilitating speciation[7]. In *D. melanogaster*, ten genes of the spiracle gene network are required for the formation of the posterior lobe, with their activation in at least seven cases being regulated in both structures by the same *cis*-regulatory elements (CRE). The study of two of these enhancers, revealed that the same DNA-binding sites activate the CRE's expression in both organs, making this one of the best-characterised cases of whole gene network co-option[3].

The recruitment of a gene network to a new organ exposes it to different selective pressures that may accelerate the appearance of novelties[8]. One such novelty is the expression of the posterior segment determinant *engrailed* in the anterior compartment cells of the A8 segment[5]. Engrailed is crucial during segmentation, and its expression in the anterior compartment is surprising given that En has been localised to the posterior cells all along arthropod evolution[9–12].

*Drosophila* segmentation results from the activation of the segment-polarity genes *engrailed* (*en*), *hedgehog* (*hh*) and *wingless* (*wg*) in periodic stripes of cells along the antero-posterior axis of the embryo[13–15]. Once activated, the segment-polarity genes engage in cross-regulatory interactions that maintain their expression. As a result, En and its direct target *hh* become expressed throughout development in the posterior compartment of every segment, where they regulate cell tension and adhesive characteristics that prevent posterior cells from mixing with anterior compartment cells, generating stable signalling boundaries[16]. Mutant embryos for either *en* or *hh* result in an almost complete fusion of segments. As the posterior spiracle is one of the few circumferential organs in the embryo, *en* activity in anterior A8 cells could be required for establishing the circumferential information pattern necessary for spiracle organogenesis that contributed to the evolution of the protruding posterior spiracles characteristic of dipteran larvae.

Here, we investigate when *en* was recruited to the anterior A8 cells (A8a) and the *cis* and *trans* regulatory elements responsible for it. We find that A8a expression appeared in Diptera before the evolution of the posterior lobe and present evidence that this is associated with a previous posterior spiracle gene network co-option event to the testis mesoderm. Working with *Drosophila melanogaster*, we show that Engrailed expression in the A8a compartment is not required for the spiracle's development and that the *engrailed* CRE controlling spiracle expression is required in the testis cyst cells for spermiation. Our work presents an example of repeated sequential gene network co-option events involving tissues of different germ layers, and shows how this resulted in the generation of a bona fide pre-adaptive developmental expression novelty: the activation of the En transcription factor in the anterior compartment of the embryonic A8 segment where, despite having no specific function, it opens the possibility of this important developmental factor acquiring one in the future. We show that the expression of *en* in the anterior compartment of the A8 segment, was likely caused by the regulatory interlocking of the co-opted networks. We propose gene network interlocking occurs as the result of the use of the same gene network in several organs, so that any change to the network because of its functionality in one organ, will be mirrored in all organs even if it has no selective advantage in some of them.

## Results

Engrailed expression in the posterior part of the segment has been conserved in arthropods for over 500 million years[9–12]. Despite such wide conservation, some exceptions have been described in *D. melanogaster* where *en* becomes activated in anterior cells of the wing primordium and in the posterior spiracles[5,17]. We have found that although *hh* and *en* are initially expressed in a coincident stripe of A8 posterior cells (Fig. 1c-c"), after stage 11, En expression in A8 diverges from that of *hh*, being turned off from the ventral posterior cells and activated in dorsal anterior cells (Fig. 1d). This pattern re-specification generates a circumference of En expressing cells around the spiracle opening that could be important for stigmatophore morphogenesis.

To infer when this expression novelty appeared we stained different Diptera species using the cross-reactive antibodies anti-Sal, which serves as a marker for the stigmatophore, and anti-En. For this purpose, we analysed three cyclorrhaphan species with similar larval body plans (Supplementary Fig. 1). En and Sal expression patterns are almost indistinguishable between *Drosophila melanogaster* and *Drosophila virilis*, which diverged about 40 million years ago (Sophophora Drosophila) (Fig. 1e, f). In contrast, while Sal is expressed in the stigmatophore of *Episyrphus balteatus*, which diverged from *Drosophila* about 100 million years ago[18], En does not form a ring around the spiracle opening, being expressed as a stripe (Fig. 1g). This stripe is very similar to that formed in the A9 segment suggesting that En expression in *Episyrphus* is restricted to the posterior compartment and that expression of En in A8a has been recently acquired in the brachiceran diptera. Comparison of *Episyrphus* and *Drosophila* larvae shows that the stigmatophore of *E. balteatus* is noticeably less protrusive than that of *D. melanogaster* or *D. virilis* (Fig. 1h–j) suggesting that En recruitment to the A8a compartment may be responsible for this organ shape difference.

### Regulation of *engrailed* anterior compartment expression

To understand the genetic and molecular mechanisms driving activation of *en* in the anterior compartment, we searched in *D. melanogaster* for the specific CRE activating its expression in A8a cells and the transcriptional regulators interacting with it. Our search was facilitated by a previous analysis that identified the CREs present in the *invected-engrailed* locus[19]. Analysis of the *enH-lacZ, enM-lacZ* and *enP-lacZ* reporters driving stripe expression at different stages, reveals they are exclusively active in the posterior cells of the A8 segment (Supplementary Fig. 2a–d). Two reporters are specifically expressed in A8: the *enX-lacZ* reporter is active in small groups of isolated cells that probably represent spiracle sensory elements, while the *enD-lacZ* reporter shows expression in a ring of cells surrounding the spiracle's opening (Supplementary Fig. 2e–f).

Dissection of *enD* localised the posterior spiracle-specific enhancer to a 439 bp region (*enD0.4*, Supplementary Fig. 3). The activation of reporter genes containing this element appears first in a dorsal stripe in the anterior compartment of A8 (Supplementary Fig. 3c–e) that expands at later stages to surround the spiracle opening coinciding with En protein expression in the stigmatophore (Fig. 2a). As all reporters containing these regulatory elements drive similar expression (Supplementary Fig. 3f–g), in what follows we use them interchangeably to study *en* regulation in the posterior spiracle using either *enD-lacZ, enD-ds-GFP*, or *enD-0.4-mCherry* as specified in each particular figure.

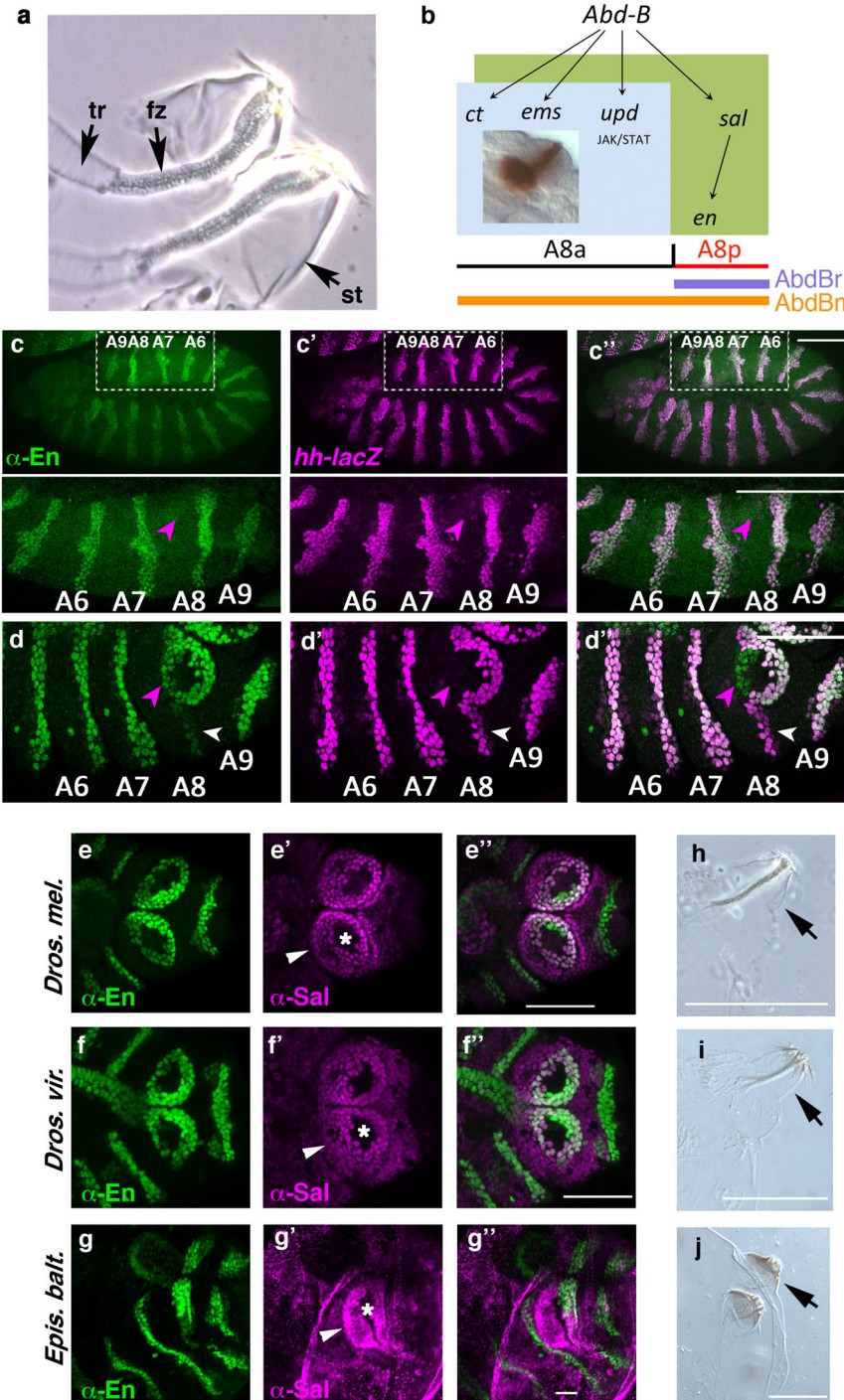

**Fig. 1 | Evolution and dynamic expression of *en* in the anterior A8 segment.**
**a** Close-up of the larval posterior spiracles showing the internal filters (fz) formed by the spiracular chamber cells, lodged in the protruding stigmatophores (st) and connecting to the tracheae (tr). **b** Scheme of the st11 early posterior spiracle organogenetic gene network activated in A8 by Abd-B. Light blue represents the spiracular chamber primordium that eventually will invaginate to form at st16 the fz (brown, inset). Olive green represents the cells of the stigmatophore primordium. The spatial expression domain of the Abd-B m (orange line) and Abd-Br (mauve line) isoforms and the location of the antero-posterior A8 compartment boundary are indicated. All spiracular chamber cells form in the Abd-Bm expressing anterior compartment, while the stigmatophore primordium comprises both anterior and posterior dorsal A8 cells. **c**, **d** *hh-lacZ* embryos double stained with anti-En (green) and anti-ßGal (magenta) at progressively later stages: (**c-c''**) st11, (**d-d''**) st14. Magenta arrowheads point to anterior A8 cells gaining En expression at st14, white

arrowheads point at the A8 posterior cells losing En expression. The panel to the right shows both channels. To facilitate segment comparison between the extended germ band embryo (**c**) and the retracted germ band embryo (**d**), the close-up of the squared region shown in panel c, has been rotated 180°. The location of the posterior abdominal (A6–A9) compartments is labelled. **e**–**g** Wild-type embryos double stained with anti-Sal (magenta) and anti-En (green) in *Drosophila melanogaster* (**e-e''**), *Drosophila virilis* (**f-f''**) and *Episyrphus balteatus* (**g-g''**). White arrowheads in (**e'-g'**) point at the stigmatophore Sal expression and asterisks mark the spiracle opening. Note that the cross-reactive Sal antibody staining has some background (**g'**) due to unspecific binding to the vitelline membrane that cannot be completely removed. **h**–**j** Cuticles of recently hatched *Drosophila melanogaster* (**h**), *Drosophila virilis* (**i**) and *Episyrphus balteatus* larvae (**j**) with arrows pointing to the protruding stigmatophore. Scale bars: 50 μm in **c**–**g** and 100 μm in **h**–**j**.

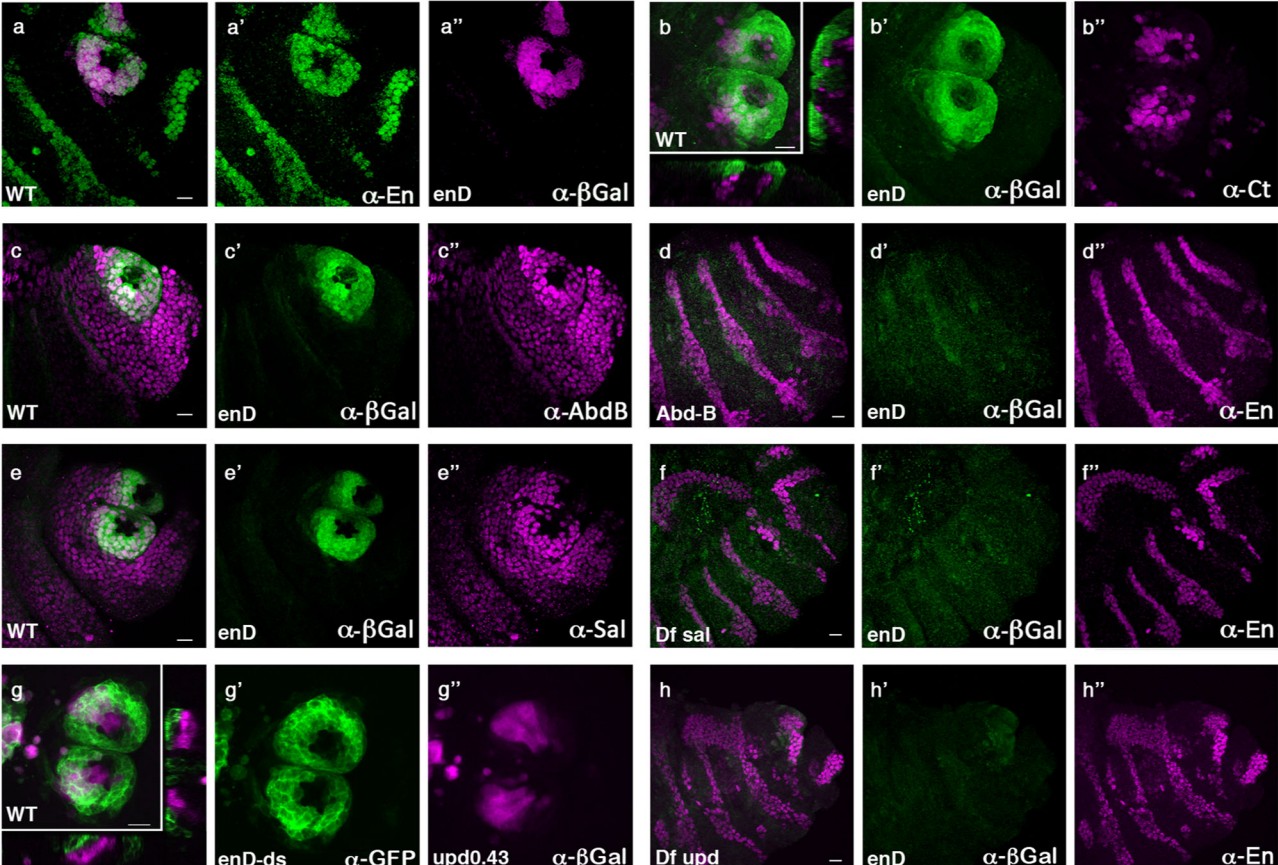

**Fig. 2 | Regulation of En expression in the stigmatophore of *Drosophila melanogaster*.** Expression of *enD* enhancer in wild type (**a**–**c**, **e**, **g**) or posterior spiracle mutant embryos (**d**, **f**, **h**). **a** *enD-lacZ* embryo double stained with anti-En (**a'** green) and anti-ßGal (**a"** magenta). **b**, **c e** *enD-lacZ* embryos double stained with anti-ßGal (green) and either anti-Ct (**b**), anti-AbdB (**c**), or anti-Sal (**e**) in magenta. **g** *enD-GFP upd0.43-lacZ* embryo double stained with anti-GFP (**g'** green) and anti-ßGal (**g"** magenta). **d**, **f**, **h** *enD-lacZ* embryos homozygous mutant for either *Abd-B^{MI}* (**d**), *Df(2) 5 sal-salr* (**f**) or the *upd* locus deficiency *Df(1)os1A* (**h**), double stained with anti-ßGal (green) and anti-En (magenta). Panels **b** and **g** also show orthogonal sections to demonstrate Ct and *upd0.43* are expressed in the internal spiracular chamber cells and *enD-lacZ* in the external stigmatophore cells. All embryos are st14. Scale bar: 20 µm.

To identify the upstream regulators controlling *engrailed* expression in the anterior compartment we analysed the reporter's activation in mutants for the segment polarity and for the posterior spiracle gene networks. Expression of *enD-lacZ* in mutants for either *en^E*, *hh^{AC}* or *wg^{CX4}* homozygous embryos is maintained in the posterior spiracles despite their global abnormal morphology (Supplementary Fig. 4a–c), suggesting the enhancer is not under the direct regulation of the segment-polarity gene network.

We then determined the spatial relationship of spiracle cells activating the *enD* enhancer with respect to various posterior spiracle gene network regulators[6,20,21] and found they belong to the protruding stigmatophore (Fig. 2b, c, e, g).

Analysis of posterior spiracle gene network mutants shows the *enD* reporters are not expressed in *Abd-B^{MI}* nor in *sal* null mutants (Fig. 2d, f) or in embryos carrying the *Df(1)os1A* deletion that lack the Upd, Upd2 and Upd3 ligands activating the JAK/STAT signalling pathway (Fig. 2h). Mutations in other posterior spiracle genes tested do not affect *enD* expression despite having abnormal spiracle development (Supplementary Fig. 5a, b).

As *upd* is not transcribed in the stigmatophore, these results suggest that Upd diffusion from the spiracular chamber primordium activates the JAK/STAT receptor in the adjacent Sal expressing stigmatophore cells. We tested this model by expressing ectopically in all ectodermal cells either *UAS-upd* or *UAS-sal* with the pan-ectodermal driver *69B-Gal4* line. Ectopic Upd expression results in the expansion of *enD*-reporter expression to all dorsal A8 and A9a Sal-expressing cells

(Supplementary Fig. 5c). In contrast, ectopic expression of Sal in the whole ectoderm does not modify *enD* activation in A8 (Supplementary Fig. 5d). To test if simultaneous Sal and Upd activation of JAK/STAT would suffice to activate *enD* expression in the dorsal ectoderm, we simultaneously expressed *UAS-sal* and *UAS-upd*. Although the co-expression of both proteins results in additional ectoderm expression, this only appears in the posterior A7 segment (Supplementary Fig. 5e) suggesting that Abd-B or another posterior abdominal factor is also required for *enD* activation.

## Regulation of *enD* expression by STAT and Abd-B

To find out if the *enD* CRE is directly regulated by Abd-B, Sal or STAT, we searched the region bioinformatically to identify putative DNA-binding sites (Supplementary Fig. 6a).

Using electro mobility shift assays (EMSA), we observed that activated STAT binds to an oligo containing the TTC(4n)GAA STAT-binding site[22]. Mutation of this sequence prevents activated STAT binding to the oligo (Supplementary Fig. 6b). Similarly, we observed full-length Abd-B binding to oligos containing its putative DNA-binding sites (Supplementary Fig. 6c, lane 6). Binding specificity was confirmed by the super-shift generated when adding anti-Abd-B antibody (lane 7) and by the lack of binding of an Abd-B protein with a mutation affecting Asn51 amino acid in the DNA-binding homeodomain (lanes 8 and 9). We failed to confirm direct Sal protein binding in EMSA experiments using the Sal zinc-finger domain.

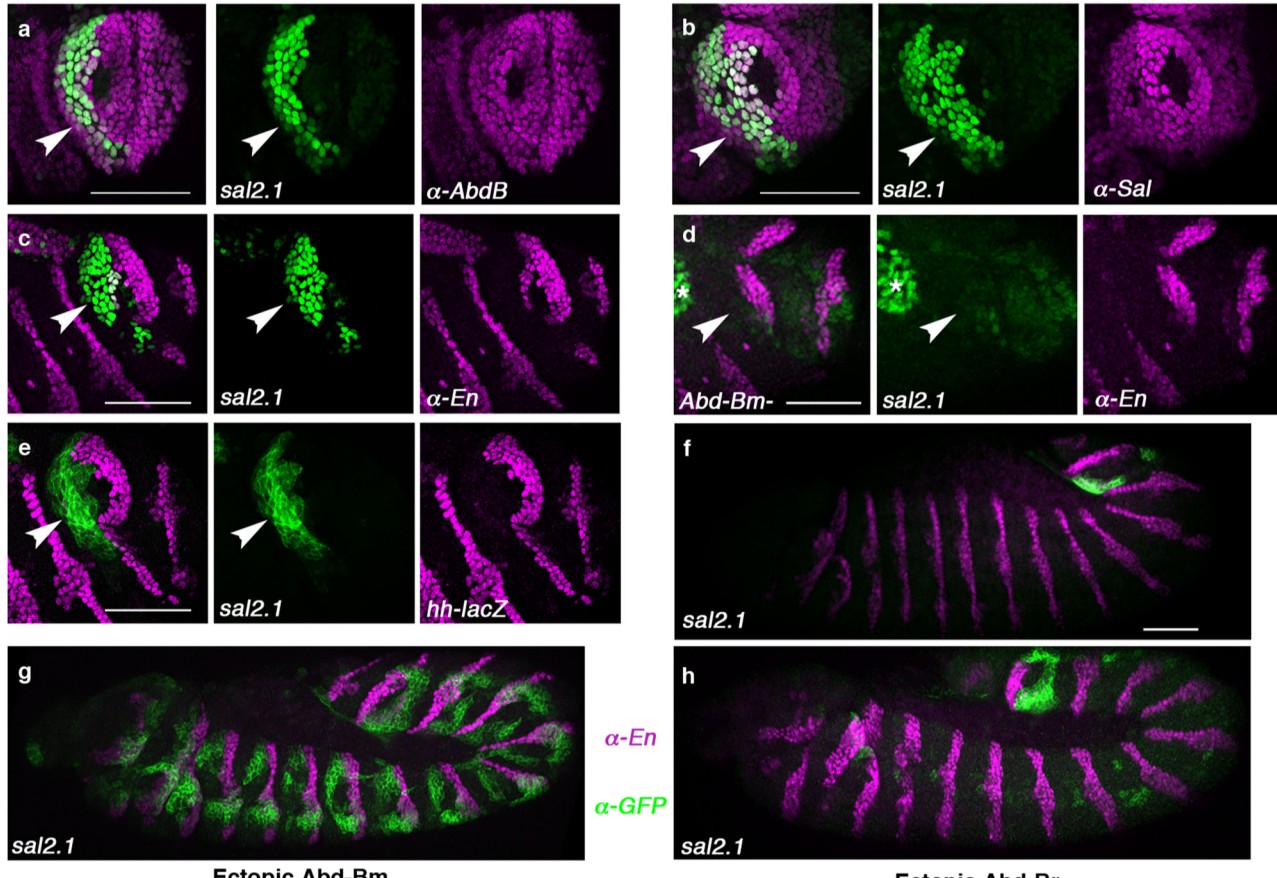

**Fig. 3 | Expression and regulation of *sal2.1* in the A8 segment. a–c** Expression of *sal2.1-lacZ* in wild type st13 embryos double stained with anti-ßGal (green, arrowheads) and anti-AbdB (**a**), anti-Sal (**b**), or anti-En (**c**) in magenta. **d** Expression of *sal2.1-lacZ* in *Abd-B^MS* mutant embryos lacking the m isoform. **e** Expression of *sal2.1-PH-GFP* in *hh-lacZ* embryos stained with anti-GFP (green) and anti-ßGal (magenta).

**f** Expression of *sal2.1-PH-GFP* (green) in wild type embryos stained with anti-En (magenta). **g, h** Ectopic activation with the ectoderm-specific line *69B-Gal4* of the *UAS-Abd-Bm* (**g**) or *UAS-Abd-Br* isoform (**h**) in *sal2.1-PH-GFP* embryos stained with anti-GFP (green) and anti-En (magenta). Arrowheads in **a–e** point at anterior A8 cells, asterisk in panel **d** indicates hindgut expression. Scale bar: 50 μm.

We tested the in vivo relevance of the biochemically identified sites on the *enD0.4* reporter element. Mutation of the 4n TTCCAGC-GAA STAT-binding site in *enD0.4* to TTCCAGCGtt in *enD0.4StatMut*, decreased reporter activation in the posterior spiracle at the early 11–12 stages (Supplementary Fig. 6d, e). However, *enD0.4StatMut* spiracle expression reappears at stage 14, indicating that this reporter element is also controlled by late spiracle gene network elements independently of STAT. Mutation of both Abd-B putative sites [site 260 CCCATAAAAAT to CCCAcggAAAT and site 359 GGATTTATGGC to GGATccgTGGC] in *enD0.4AbdB260-359Mut* completely abolished reporter expression (Supplementary Fig. 6f). These results confirm the direct regulation of *en* in the posterior spiracles by Abd-B and STAT.

### Regulation of *enD* expression by Sal

Sal is crucial for *enD* expression and previous work showed *sal* is the earliest gene expressed in a circumferential stigmatophore-specific pattern[4] with spatio-temporal dynamics of expression preceding *enD* reporter expression. Thus, to understand how the gene network controlling stigmatophore organogenesis has been established, it is relevant to know how *sal* spiracle transcription is controlled.

The *sal* locus spans over 110kbp between the *salr* and the *sal* genes[23]. To identify a *sal* CRE potentially active in the posterior spiracles, we searched bioinformatically for regions containing putative binding sites for spiracle cascade regulators. We identified a 1,8kbp fragment (*sal2.1*) driving reporter expression in the stigmatophore overlapping with Abd-B, Sal and anterior En expression (Fig. 3a–c).

Interestingly, a comparison with a *hh-lacZ^P30* enhancer trap line[24] shows *sal2.1* is expressed exclusively in the anterior A8 compartment (Fig. 3e).

To identify *sal2.1* regulators we analysed its expression in posterior spiracle mutants. The *Abd-B* gene gives rise to two isoforms with different spatial patterns of expression (Fig. 1b). The Abd-Br isoform is expressed from A8p to A10, while Abd-Bm is expressed from A5-A10[25]. Thus Abd-Bm is the only isoform expressed in A8a. We compared the regulatory capability of each isoform by ectopically expressing them in the ectoderm with the *69B-Gal4* line. Abd-Bm induces ectopic *sal2.1* activation in anterior trunk segments while Abd-Br does not result in strong *sal2.1* ectopic activation (Fig. 3f–h) suggesting an isoform-specific regulation. Next, we analysed *Abd-B^MS* mutant embryos, which lack exclusively the Abd-Bm isoform function[25,26], and found *sal2.1* expression disappears from anterior A8 (Fig. 3b, d). Mutations in other posterior spiracle genes tested do not affect *sal2.1* expression despite having abnormal spiracle development (Supplementary Fig. 5f–i).

Taken together, our results suggest that the novel A8a segment-specific activation of En required the function of at least two Abd-B regulated enhancers: *sal2.1* inducing Sal expression in A8a at stage 11 and *enD* integrating the Abd-B, Sal and JAK/STAT signalling pathway inputs.

### Analysis of *enD* enhancer function

To find out what is *enD* required for during development, we generated *enDΔ*, a deletion in the endogenous gene encompassing the

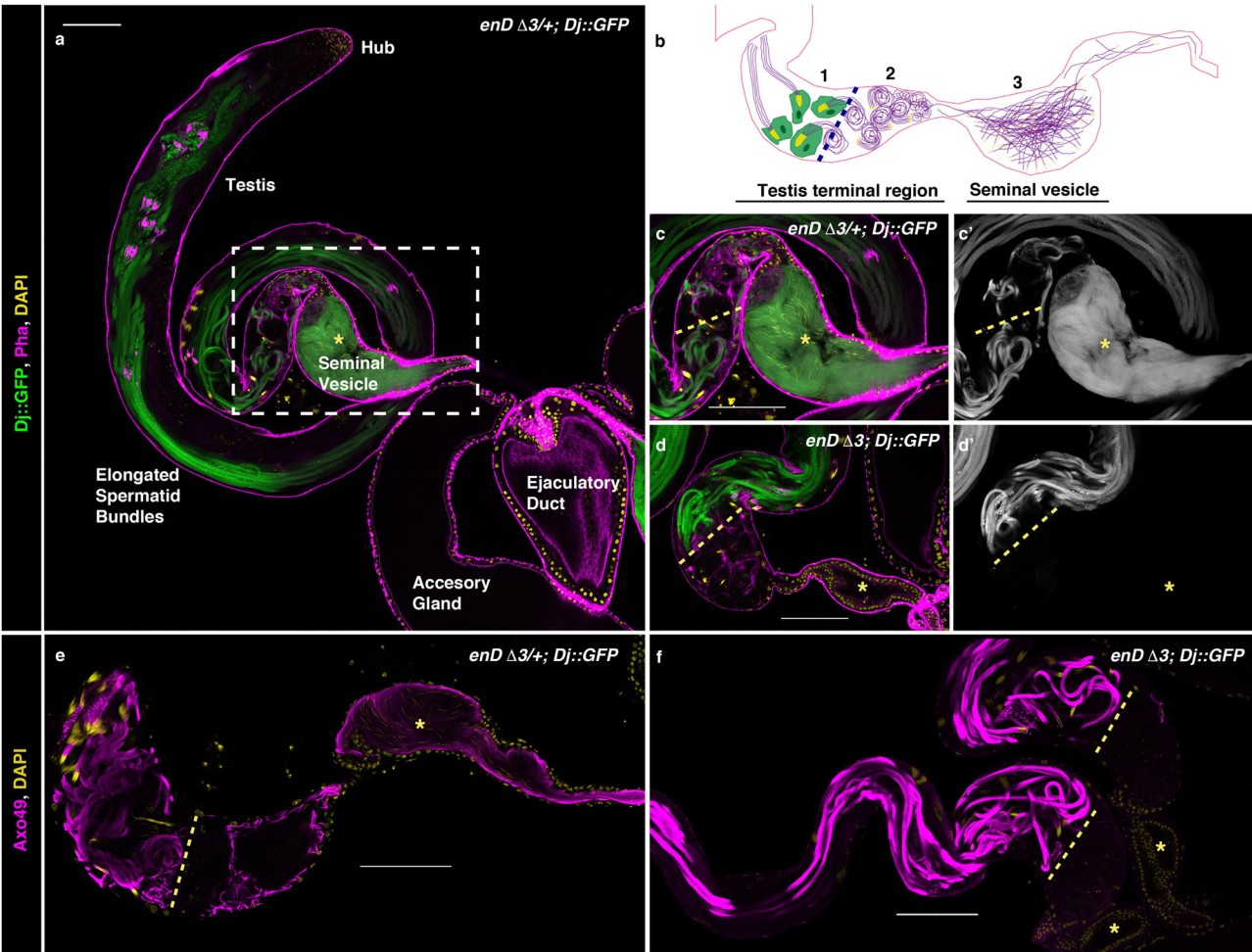

**Fig. 4 | Defective sperm release in *enDΔ* mutant testis.** Five-day-old adult testis with the sperm tails labelled with Dj::GFP (green, or grey) in control heterozygous (**a**, **c**) or homozygous *enDΔ* males (**d**). **b** Schematic drawing of the *Drosophila melanogaster* wild-type testis terminal region indicating (1) the Head Cyst Cells (HCC, green) tightly holding the sperm heads (yellow); (2) the coiled sperms and (3) the sperms after their release to the seminal vesicle. **c** Close-up of the squared region in **a**. In heterozygous males (**a**, **c**) spermiation leads to sperm accumulation in the seminal vesicle (asterisk). In homozygous *enDΔ* males (**d**) sperms cannot be detected in the seminal vesicle (asterisk) nor in the distal part of the terminal testis region (right of the discontinuous yellow line). **e**, **f** Testes with the sperm tails labelled with Axo49 (magenta) in the testis terminal region and seminal vesicle (asterisk) of control heterozygous (**e**) or *enDΔ* homozygous (**f**) males. All testes are counter-stained with DAPI (yellow). In (**a**, **c**, **d**) filamentous Actin is labelled with Rhodamine phalloidin (magenta). Scale bar: 100 μm.

*enD0.4* element that eliminates En expression from the A8a compartment (Supplementary Fig. 3a, h, i and Methods). Unexpectedly, *enDΔ* homozygous mutants develop normal posterior spiracles but are male sterile.

Despite *enD* reporter being active in third instar larval and pupal genital discs (Supplementary Fig. 7a, b), the *enDΔ* adult male genitalia appear normal (Supplementary Fig. 7c–f). Quantification shows that the posterior lobe area size in *enDΔ* mutants is intermediate to that of the wild type and the *enDΔ/+* males, both of which are fertile, indicating that male sterility is not due to genital defects (Supplementary Fig. 8). To test if the sterility could be due to abnormal spermatogenesis we labelled the sperm tails with the Dj::GFP transgene[27]. In control males, *Dj::GFP* labelling is first detected in the elongated sperm tails extending along most of the testis as they individualise (Fig. 4a), and is maintained when the sperms exit into the seminal vesicle (Fig. 4b, c). In *enDΔ3; Dj::GFP* mutant testis, we only observe GFP labelling at the sperm individualisation stage, with no GFP detectable in the seminal vesicle nor at the end of the testis terminal region which only contains cyst cell remnants (Fig. 4d). The same is observed when labelling the sperm axoneme with anti-Axo49 antibody (Fig. 4e, f). These results indicate that *enD* may be required when the mature sperms abandon the testis.

### Posterior spiracle gene network expression in the testis
In *Drosophila melanogaster*, the germ cell niche is located at the testis apex in a structure known as the hub (Fig. 4a). When the germ stem cell divides, it gives rise to a sperm progenitor cell (gonialblast) that separates from the hub and becomes encapsulated by two highly specialised mesodermal cells (the cyst cells). The cyst cells do not proliferate but will protect and signal to the gonialblast as it divides[28,29]. Inside each cyst, the gonialblast experiences four mitoses and a meiosis generating 64 clonal spermatids that elongate (Fig. 4a). When the spermatids individualise to give rise to the spermatozoa, the two cyst cells differentiate to form a head cyst cell (HCC) that forms an Actin basket holding tightly the 64 sperm heads, and a tail cyst cell (TCC) that elongates to surround the growing sperm tails (Fig. 4b). After the 64 spermatozoa individualise, they coil at the testis terminal region where, during the phase known as spermiation, generate forces that lead to their liberation from the cyst and exit to the seminal vesicle (Fig. 4c)[30,31].

Because of the observed late spermatogenesis defects, we analysed *enD* activity in the adult testis. *enD* expression shows it is active at the testis terminal region in the head cyst cells (HCC), which also express low levels of En (Fig. 5a–c). We next studied if the same gene network regulates *enD* in the testis and in the spiracles. Abd-B

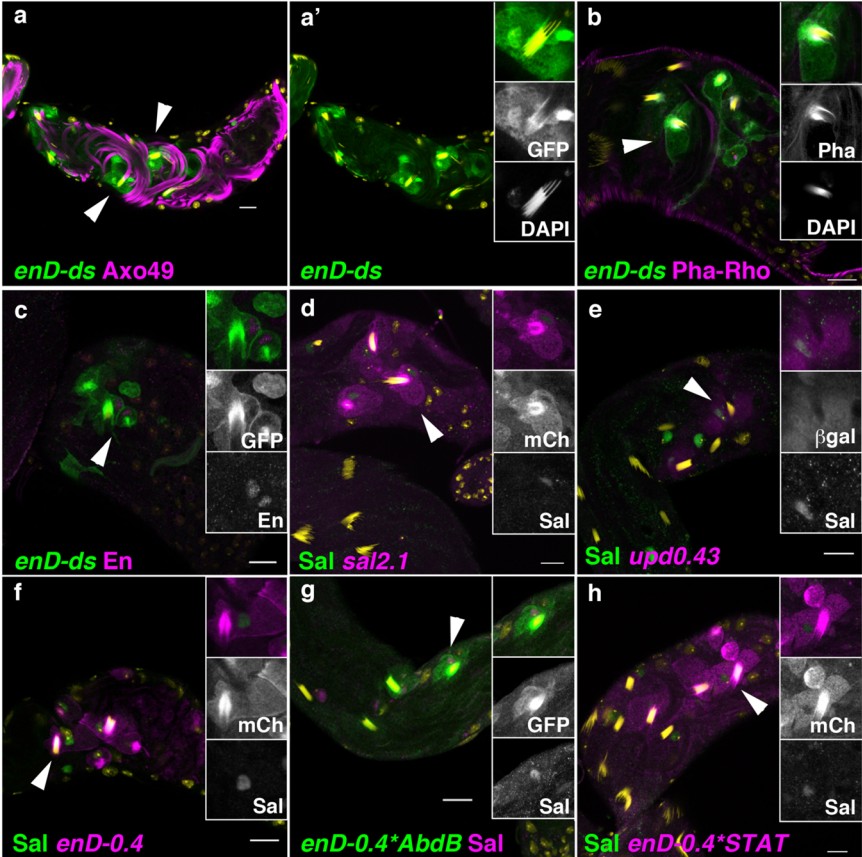

**Fig. 5 | Expression of posterior spiracle network genes in the testis terminal region. a–e** Expression of *enD*, *sal2.1* and *upd0.43* enhancers in the HCC at the testis terminal region. (**a**) *enDds-GFP* (green) is expressed in the HCCs holding the 64 sperm head bundles. Sperm tails labelled with Axo49 are shown in magenta, and sperm heads are in yellow with DAPI. (**a'**) shows only GFP and DAPI stainings and a close-up of an HCC holding the sperm heads. **b** The HCC labelled in green with *enDds-GFP* can be distinguished by the formation of an actin basket [labelled with phalloidin (magenta or white)] holding the 64 sperm head bundles (yellow).

**c** *enDds-GFP* is active in HCC cells which also express low levels of endogenous Engrailed (magenta). **d** *sal2.1-mCherry* is active in cells expressing low levels of endogenous Sal protein. **e** *upd0.43-lacZ* (magenta) is active in a subset of HCC expressing Sal (green). Wild type *enD0.4-mCherry* reporter expression in HCCs (**f**). Reporter expression is maintained in *enD0.4* when the putative Abd-B DNA-binding sites are mutated (**g**), or when the putative STAT sites are mutated (**h**). In (**a**, **b**, **d**–**h**) DAPI staining is yellow. Scale bar: 20 μm.

expression has been previously observed in the embryo's testis mesodermal somatic cells[32]. However, in the larval and adult testis, Abd-B is no longer expressed in the mesoderm cells although it is detectable in the germ cells[33]. We confirmed the lack of Abd-B expression in mesodermal testis cells, which accords with a report showing that ectopic Abd-B induction interferes with testicular development[34]. In contrast, we found expression of Sal in the HCC nuclei at the testis terminal region (Fig. 5d–f). We also found that the *upd-Gal4* enhancer trap line driving the UAS-H2AmChery-P2A-eGFP-PH reporter labels the HCC nuclei and membranes (Supplementary Fig. 9b). Furthermore, the 10xSTAT-GFP reporter, which is expressed in cells activating the JAK/STAT signalling pathway[35,36], is also active in the HCC where we can also detect nuclear STAT-GFP expressed from a BAC element (Supplementary Fig. 9c, d). To find out if the enhancers driving the expression of *sal* and *upd* in the HCC are the same as those responsible for their activation in the posterior spiracles, we tested the spiracle-specific *upd0.43-lacZ* reporter and *sal2.1-GFP*. In both cases, we observed activation in the HCC (Fig. 5d, e). Other spiracle network genes like *cv-c* and *crb* are also activated in the HCC and in the case of *crb* using the same enhancer driving spiracle expression (Supplementary Fig. 9e, f)[37] indicating that part of the posterior spiracle gene network has been co-opted to the HCC mesodermal cells.

To test the extent to which the *enD* enhancer regulation is conserved, we analysed the expression of the Abd-B and STAT mutated binding site reporters. As could be expected from the absence of Abd-

B expression in the HCC, mutating the Abd-B binding sites does not affect the testis enhancer expression (Fig. 5g). Mutation of the STAT-binding sites does not affect the construct's expression either (Fig. 5h), this may be because it still has a secondary input from the gene network as happens in the posterior spiracles (see above).

Finally, we analysed if Sal and En are expressed in the testis of *D. virilis* and *E. balteatus* (Fig. 6). We detected equivalent expression of both Sal and En in the testis terminal region of *D. virilis* (Fig. 6c, d) but not in *E. balteatus* (Fig. 6e, f), suggesting these genes were co-opted to the adult testes in the *Drosophilids* after they diverged from *Episyrphus* ancestors. In fact, *Drosophila* and *Episyrphus* testis morphology is very different and even the sperm of *Episyrphus balteatus* have shorter tails as shown by staining the axoneme with Axo49 antibody (Fig. 6b, d, f), suggesting that, among other factors, the posterior spiracle gene network co-option may have contributed to their morphological divergence.

## Discussion

Genetic co-option is an evolutionary source of novelty[1,8,38–41]. Here we studied how the expression of the posterior compartment determinant Engrailed in anterior cells, was acquired during the organogenesis of the posterior spiracles. Unexpectedly, we found this novelty is required for spermiation, the process by which the sperms are released from the somatic cyst cells encapsulating them, but is functionally irrelevant for the development of the larval posterior spiracles.

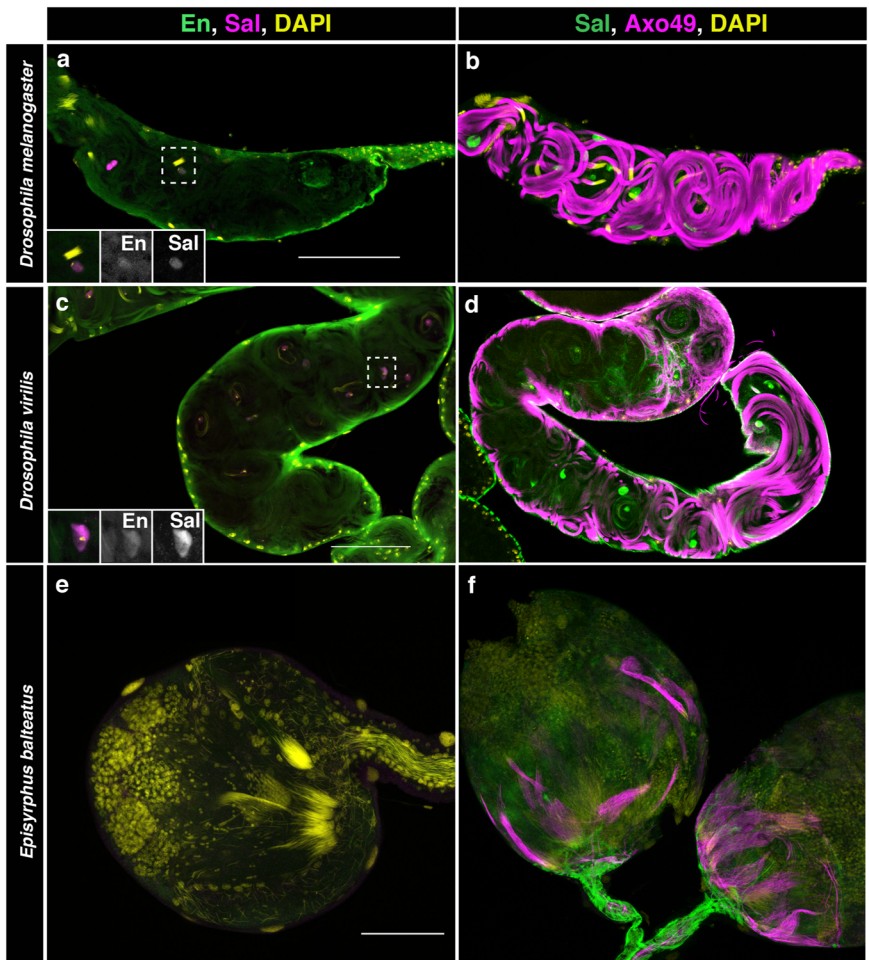

**Fig. 6 | Expression of Sal and En in the testis of three dipteran species.** Testis terminal region of *D. melanogaster* (**a**, **b**), *D. virilis* (**c**, **d**) and whole testis of *E. balteatus* (**e**, **f**) stained with anti-Engrailed and anti-Spalt (**a**, **c**, **e**) or anti-Spalt, and the axoneme marker anti-Axo49 (**b**, **d**, **f**). *D. virilis* testes are larger but similar in shape to those of *D. melanogaster* and have a similar expression of En and Sal in the terminal region (**a**, **c**) where the sperms coil and individualise (**b**, **d**). The testis morphology of *E. balteatus* is completely different and expresses neither En nor Sal (**e**). Axoneme staining with anti-Axo49 and DAPI reveals the formation of shorter sperms in *E. balteatus* that do not coil (**f**). DAPI DNA staining in yellow. Scale bar: 100 μm.

Previous work showed that the ectodermally expressed posterior spiracle gene network had been co-opted to the ectodermal male genitalia in the *Drosophila melanogaster* clade[3]. Here, we show the same gene network has also been co-opted to the mesodermal head cyst cells in *Drosophila*. We found that part of the posterior spiracle gene network becomes activated in the HCCs using the same enhancers driving expression in the posterior spiracles.

Although Abd-B is expressed in the somatic testis cells during embryogenesis, it is not expressed in the adult HCCs. We suggest two alternative explanations. The first possibility is that one of the spiracle primary targets, many of which encode transcription factors, becomes expressed in the testis HCC independently of Abd-B regulation, and this results in the activation of the other spiracle genes due to cross-regulatory network interactions. Alternatively, the embryonic Abd-B expression could be speculated to epigenetically modify the spiracle gene network leaving it in a poised state that later could become activated in the absence of Abd-B.

The finding that the posterior spiracle gene network has been co-opted twice to different organs, suggests that the coordinated activation of several transcription factors and signalling molecules of the network does not necessarily cause detrimental effects. The reported effects of experimentally inducing the ectopic activation of the eye gene regulatory network in *D. melanogaster*, a situation akin to what may happen during co-option, may help in understanding this. *eyeless*

activation in the imaginal primordia results in the formation of ectopic eyes and causes morphological alterations leading to lethality[42]. However, it has been noted that ectopic Eyeless expression in the wing only induces eye development in proximal cells where Dpp is also expressed, and similar results were observed with Hh[43,44] showing that the activation of a gene network inducer does not cause developmental transformations in all cells where it is expressed. Thus, when the co-option of a gene network causes developmental transformations reducing the animal's fitness, it will be lost. However, if the co-option had no influence on local developmental processes, it could be tolerated giving the opportunity to the gene network elements to interact with allelic variants present in the population whose interaction could result in selective pressures fixing the trait. Such a series of steps may have led to the co-option of other developmental gene networks such as the activation of the appendage gene network in the butterfly wings that resulted in the formation of novel wing spot patterns[2,45].

Expression of Engrailed in posterior metameric stripes is characteristic of arthropods, but is also observed in Onycophorans and in certain worms indicating an ancient origin[12,46,47]. In flies, anterior En activation has rarely been reported[48], and in *D. melanogaster*, anterior expression is the exception. En activation in A8a associated to the posterior spiracle is present in *D. virilis*, but not in *E. balteatus*, suggesting it originated in the higher Diptera (Brachicera). This is

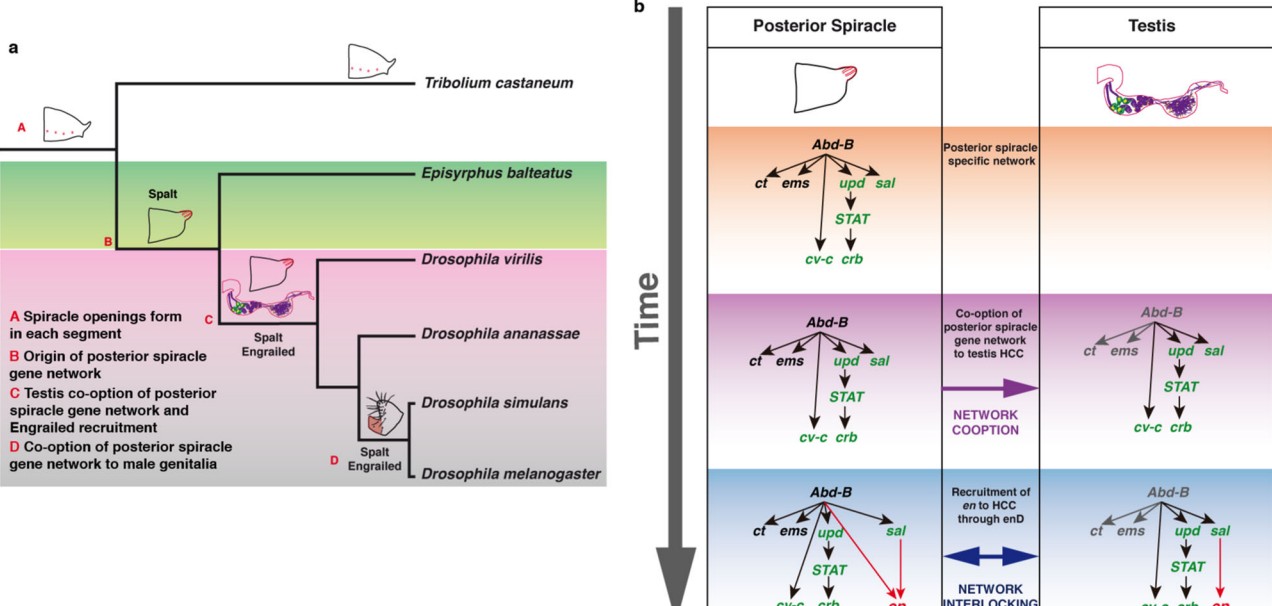

**Fig. 7 | Interlocking of co-opted gene networks. a** Simplified evolutionary tree representing the consecutive co-option events that sequentially recruited the posterior spiracle gene network to the testis and the male genital posterior lobe in *Drosophila melanogaster*. Temporal lines not to scale. **b** Schematic representation of the co-opted posterior spiracle gene network elements (green) to the testis HCC and the posterior recruitment of Engrailed function (red) to the testis that resulted in its simultaneous aphenotypic recruitment to the posterior spiracles caused by the networks' interlocking. Genes highlighted in grey are not expressed in HCC.

supported by analyses showing that in other Diptera like *Bactrocera dorsalis* (*Tephritidae*) and *Lucilia sericata* (*Calliphoridae*), that present well-developed larval posterior spiracles, *engrailed* expression is restricted to the posterior A8 segment[49,50]. We have found that A8a *engrailed* spiracle expression does not depend on the segmentation gene network, but is regulated by the posterior spiracle network through Abd-B, Sal and the JAK/STAT signalling pathway. Despite its complex regulation, this *en* enhancer has no function in *Drosophila melanogaster*'s spiracle organogenesis. Taken together, Engrailed expression in the anterior compartment of A8 was acquired after functional posterior spiracles already existed in dipteran larvae and its origin had to do with testis evolution. Future experiments should establish if this occurred due to the de novo appearance of the *enD* CRE, or to a silent *enD* cryptic enhancer becoming activated due to the recruitment of a new transregulatory factor in the spiracles/testis gene network or to changes in chromatin accessibility.

Most insects have a pair of non-protruding spiracle openings in each trunk segment (Fig. 7a branch A). In the hemimetabolous bug *Oncopeltus fasciatus* nymph or the holometabolous beetle *Tribolium castaneum* larva these are formed by an internal spiracular chamber that expresses the Ct protein[51,52], however, in *Oncopeltus*, Ct expression is not regulated by Hox proteins as it is in *Drosophila*, but depends on the tracheal protein Trachealess. Although in *Tribolium*, Spalt is expressed in the lateral ectoderm, it is not associated with the spiracles. Similarly, Engrailed is restricted to a posterior stripe of cells in the segment that does not surround the spiracular opening[53]. Our analysis of *E. balteatus* embryos suggests Sal was recruited in Diptera to the protruding stigmatophore before *engrailed* was expressed in A8a (Fig. 7a branch B). Analysis of Sal and En expression does not reveal any activation in *E. balteatus* testis, indicating the spiracle network is not expressed in the gonads of all Diptera. This situation changes in *Drosophila* species where both Sal and En are expressed in A8a associated with the spiracles as well as in the testes (Fig. 7a branch C). The co-option of the posterior spiracle cascade to the testis and the recruitment of Engrailed expression may have been the result of the major sperm size and gonad morphology changes occurring at some point

after the divergence of *Episyrphus* and *Drosophila* that could have required the selection of new genetic variants to maintain fertility (Fig. 7b purple band). As the *enD* enhancer has no apparent function in the posterior spiracles, but is required for male spermiation in *D. melanogaster*, we propose that originally this CRE was selected for its testicular function and that expression in the posterior spiracles was a side effect caused by the interlocked regulation of the co-opted gene networks in both organs which was not eliminated because it had no deleterious effects (Fig. 7b blue band). Finally, after the divergence from the *virilis* group, a second co-option event of the spiracle gene network in the male genital disc of *D. melanogaster* and closely related species led to the evolution of the posterior lobe (Fig. 7a branch D). Given the extreme variability of the posterior lobe sizes between various *D. melanogaster* genotypes it is unclear whether the *enD* enhancer has a direct function on lobe development or if its expression there is aphenotypic, and only the result of its interlocking with other functional elements of the co-opted gene network.

Our results suggest that the selection of a new trait in a co-opted gene network in one organ, results in its activation in all organs where the network is expressed. This implies a slower acquisition of new traits in co-opted gene networks as novelties would become discarded if they had detrimental effects in any of the interlocked organs. On the other hand, if the new trait (for example the activation of a novel transcription factor) was selected for its function in one organ, this would lead to the fixation of non-functional pre-adaptive traits with functional potential in all interlocked organs. Thus, we propose that co-option can lead to *sensu stricto* pre-adaptation cases, as opposed to the associated term of exaptation. While in exaptation the co-opted character had a previous selective function that has been recruited to perform a novel function (i.e., heat shock proteins recruited to form the eye crystallin, or feathers that could have served for heat regulation or sexual display before being co-opted for flight) cases like the expression of Engrailed in the posterior spiracle provide no selective advantage but could conceivably acquire it in the future, as has happened in the anterior wing of several Diptera where En has acquired a new role in wing pigmentation[48].

## Methods

### Fly stocks

Wild type flies: *Drosophila melanogaster* Oregon-R and *Drosophila virilis* were reared using standard *Drosophila melanogaster* fly food. An *Episyrphus balteatus* colony was maintained at $22 \pm 2\,°C$ in the laboratory with a 12:12 h light:dark photoperiod. Larvae were fed with aphids maintained in infested broad bean plants, while adults were fed with pollen and a 30% honey solution. Eggs were collected every 24 h and dechorionated and fixed using *Drosophila melanogaster* standard protocols.

The following *D. melanogaster* mutant stocks were used: *Abd-B$^{M1}$* null mutant for both Abd-B isoforms and *Abd-B$^{MS}$* mutant for the Abd-Bm isoform[26], *Df(1)os1A* deficient for the three Unpaired ligands (*upd1*, *upd2* and *upd3*)[54], *Df (2L)32FP-5* deficient for *spalt* and *sal-r* genes[23] (Barrio et al., 1999), *hh$^{AC}$* (BL-1749), *wg$^{CX4}$* (BL-2980) null, *Df(2R) en$^E$* (BL-2216) deficient for *engrailed* and *invected* genes[55], *Df(3L)H99* (BL-1576) deficient for *reaper*, *grim*, and *hid*[56–58], *Df(2L)drmP2* deficiency for *drm*, *sob* and *odd*[59], *ct$^{db760}$*, *ems$^{9H8361}$*, *grn$^{7j86}$* and *grn$^{7L12}$* null mutants[62].

Reporter lines: *hhP3O-lacZ* (BL-5530)[63], *enD e702−lacZ, enH sk11a-lacZ, enM slk5-lacZ, enX sn2-lacZ*[19]; *Dj::GFP*[27]. *10xSTAT-GFP* reporter[35], STAT92E-GFP BAC[BAC CH321-73F24 P[acman][64], *crb518-lacZ*[37], *cv-c::GFP*[65].

Gal4 and UAS lines: *69B-Gal4*[66], *UAS-upd*[67]; *UAS-AbdBm 1,1*[68]; *UAS-AbdBr*[22], *UAS-AbdBAsn51*[69], UAS-H2AmChery-P2A-eGFP-PH[70].

### EMSA assays

We searched bioinformatically for Abd-B, Sal and STAT putative binding sites in *enD0.4* using JASPAR (https://jaspar.genereg.net/). To test the physical binding capacity of Abd-B, Sal or STAT to the *enD0.4* DNA fragment, the *enD0.4* DNA fragment was subdivided into six overlapping fragments. Each fragment consisted of two radioactively labelled antiparallel annealed oligos. Oligo sequences are listed in Supplementary Table 1. Labelling, protein expression and binding reaction as well as gel running and imaging were performed as described[37]. Once the binding sites in the larger fragments were identified, we confirmed binding using smaller fragments of oligos 2, 4 and 6. Mutations of the putative STAT or Abd-B binding sites in these smaller fragments were used to test direct binding in EMSA. Oligos are listed in Supplementary Table 1. Protein extracts for binding reaction were obtained as described at Pinto et al.[37]. Briefly, we transiently transfected S2R+ cells (DGRC Stock 150; https://dgrc.bio.indiana.edu/stock/150; RRID:CVCL_Z831) with the following plasmids using Effectene Transfection Reagent (Qiagen # 301425): For STAT expression, pAC-GAL4, pUASt-STAT-GFP, pAC-HopTML; for Abd-B expression, pAC-GAL4 and pUASt-HA-Abd-B; for Abd-B mutant on Asn51, pAC-GAL4 and pUASt-Abd-B mut Asn51[69]. All cDNAs used encode full-length protein sequence. Transfected cells were incubated at 20 °C, harvested 72 h later and lysed using lysis buffer (10 mM HEPES pH7.0; 10 mM KCl; 5 mM MgCl2; 1% Triton X-100; 50% Glycerol; 1 mM DTT; 10uM Na-O-Ortovanadate; 1 mM PMSF). Crude extract from these cells was added directly to EMSA assays.

### CRISPR-Cas9 deletions

Three independent *enDΔ* deletions removing a 2kb-region [located at 2 R:11516500−11518500 (Flybase release: r6.45)] were induced by Genetivision. The deletions were induced using the sgRNAs enD gRNA1 and enD gRNA2 flanking enD (Supplementary Table 1) after inserting a 3xP3-GFP cassette. Sequencing showed all three deletions remove an identical region, but *enDΔ1* had a lethal off-target mutation and was discarded. Both *enDΔ2* and *enDΔ3* are viable and male sterile either when homozygous or when hemizygous over *Df(2)enE* which removes *engrailed* and *invected*.

### Constructs

*enD*[19] was subdivided into shorter fragments called *enD-ds, enD-E OL1.2* and *enD-0.4* using the primer pairs enD1for and enD3rev; enD2for and enD4rev; enDEOLfor and enDEOLrev1.2 and enDEOLfor and enD4rev respectively (Supplementary Table 1). These cis-regulatory elements were used to create *enD-ds-GFP, enD-E OL-GFP, enD0.4-mCherry* (Supplementary Fig. 3 and Supplementary Table 3).

Regulatory *sal* sequences were bioinformatically analysed to isolate regions with increased putative STAT-binding sites. Only *sal2.1*, amplified by PCR using primers Sal2.1for and Sal2.1rev (Supplementary Table 1), drove posterior spiracle expression when used as reporters (*sal2.1-lacZ* and *sal2.1-GFP*).

A 0.43 kb cis-regulatory element of *upd*, obtained amplifying with the primers upd F2 M for and upd F2 AT rev (Supplementary Table 1) was used to create *upd0.43-lacZ* which drives expression in the posterior spiracle and testis. All the above fragments were cloned into pGEMt-easy. After sequencing confirmation, fragments were excised with EcoRI and subcloned into pCASPER-NLS-LacZ, pCASPER-PH-eGFP or pCASPER-mCherry. Transgenic flies were obtained by the CBM-SO *Drosophila* transgenesis service (Spain).

In vitro mutagenesis: Putative binding sites were identified with JASPAR. *enD0.4STATmut* was made mutating the TTCCAGCGAA putative STAT-binding site in *enD0.4* to TTCCAGCGtt known to abolish STAT binding as described at[22,37]. The mutation was generated using QuikChange protocol (Stratagene) starting from pGEMt-easy:enD0.4 as a template and using the enDdwn Stat-mut for and enDdwn Stat-mut rev primers (Supplementary Table 1).

*enD0.4Abd-Bmut* was achieved using Megaprimer PCR-based mutagenesis. As template pGEMt-easy:enD0.4 was used in combination with mutagenic primers (Supplementary Table 1). Fragments containing *enD0.4WT, enD0.4STATmut* or *enD0.4Abd-Bmut* were checked by sequencing and excised from pGEMt-easy with EcoRI and cloned into pCASPER-PH-eGFP and/or pCASPER-mCherry plasmids to generate transgenic flies. AbdB binding site substitutions were made as described in refs. 22,37 as follows: Site 260 CCCATAAAAAT and site 359 GGATTTATGGC were substituted by mut260 CCCAcggAAAT and mut359 GGATccgTGGC respectively.

### In situ hybridisation

Probes were made by PCR amplification using a specific oligo followed by a second PCR reaction that introduced a T7 promoter. The QIAquick purified fragment was labelled using Digoxigenin RNA Labeling Kit (Roche). Embryos were fixed in 4% formaldehyde for 20′−30′ and hybridised at 55 °C overnight with the probe. We used anti-DIG-AP (Roche) to detect the DIG labelled probe following standard RNA in situ procedures.

### Immunohistochemistry

Embryos: *Drosophila* eggs were collected overnight at 25 °C on apple juice agar plates supplemented with fresh yeast paste. *Episyrphus* eggs were collected from the broad bean plants. All eggs were treated similarly: dechorionated 2 min in commercial bleach diluted in water (1:1), washed thoroughly and fixed for 20 min in a PBS-formaldehyde 4%/heptane mix. After removing the fixative, methanol was added and shaken vigorously a few seconds to remove the vitelline membrane. After allowing both phases to separate, sinking embryos were recovered, washed in clean methanol, rehydrated in PBS-Tween 0.1% (PBT) and preadsorbed for an hour in PBT-1%BSA at room temperature (RT). Primary antibodies were used at the described concentration, diluted in PBT-1%BSA and incubated overnight at 4 °C. Primary antibodies were washed twice for 5 min in PBT and preadsorbed 1 h in PBT-1%BSA at RT. Embryos were incubated with the secondary antibody diluted at 1:400 in PBT-1%BSA at RT for 4 h. After incubation, embryos were

washed four times 15 min in PBT at RT followed by to 2 rinses in PBS before mounting in Vectashield.

Testis: Five-day-old males were dissected in PBS and fixed 30 min at RT in 4% paraformaldehyde+0.3% Triton X-100, washed three times with PBS-Triton 0.3% and preadsorbed in PBS-Triton 0.3% + 1%BSA during 1 h. Primary antibodies were incubated overnight at 4 °C, and washed four times in PBS-Triton 0.3%. Testes were preabsorbed in PBS-Triton 0.3% + 1%BSA for 1 h at RT. Secondary antibodies diluted at 1:400 in PBS-Triton 0.3% + 1%BSA were incubated at RT for 4 h. After incubation testis were washed 4 × 15 min in PBS-Triton 0.3% at RT, rinsed twice in PBS and mounted in Vectashield.

Pupal and third larval instar genital discs were dissected in PBS. Fixation and staining protocol were performed as in embryos but with 0.1% Triton X-100.

The following primary antibodies were used (Supplementary Table 2): mouse α-Ct 2B10 1:20 (DSHB), mouse α-Abd-B 1A2E9 1:25 (DSHB), mouse anti-En 4D9 1:50 (DSHB), rabbit α-Sal (our laboratory), rat anti-RFP 1:500 (Chromotek, 5F8), mouse anti-Axo49 1:500 (Sigma-Aldrich, MABS276), mouse anti-βgal 1:1.000 (Promega, Z378A), chicken anti-βgal 1:500 (Abcam ab9361), rabbit anti-βgal 1:500 (MP Biomedicals, 8559762) rabbit anti-GFP 1:300 (Invitrogen, A11122), chicken anti-GFP 1:500 (Abcam, ab13970).

Secondary antibodies were coupled to Alexa488, Alexa555 or Alexa647 (Supplementary Table 2). Filamentous Actin was stained with Rhodamine phalloidin (Molecular Probes, R415). For RNA in situ hybridisation α-DIG-alcaline phosphatase conjugated was used (1:2000) (Roche, 11093274910).

Images were taken on an SPE or a Stellaris Leica confocal microscope making use of the company's provided software and processed using FIJI ImageJ2 (version 2.9.0/1.54f), Imaris x64 (version 8.0.2) and Adobe Photoshop CS5 (version 12.0 ×64) programs.

## Embryo cuticle preparation
Recently hatched first instar larvae were collected and mounted in Hoyer's medium. The cuticles were kept on a 75 °C plate for two days until clear.

## Posterior lobe measurements
The external genitalia of 18–28 young males (1–2 days old) were dissected and mounted in Hoyers medium. Images were taken in a Zeiss Axioplan 2 microscope at 40x magnification. Data were analysed using Microsoft Excel (version 16.16.27) and GraphPad Prism8 (version 8.4.3 (471)) (Supplementary Fig. 8, Supplementary Tables 4 and 5 and source data).

## Statistics and reproducibility
Statistical analysis in Supplementary Fig. 8. was performed as described in ref. 7. The outline of each posterior lobe was traced manually using Fiji ImageJ (ImageJ2, version 2.9.0/1.54f) and enclosed with an artificial baseline drawn in line with the lateral plate. The posterior lobe areas of different genotypes measured in pixels were compared using a one-way analysis of variance (ANOVA) with Type II Sums of Squares as recommended for unbalanced groups. Data complied with previous assumptions of normality and homoscedasticity according to Shapiro-Wilks and Levene´s tests. Subsequently, as the ANOVA results indicated significant differences between groups (ANOVA; $F_{3, 82} = 18.335$; $p$-value = $3.407*10^{-9}$) a pairwise post hoc Tukey-Kramer test was performed to assess differences between genotypes. Statistical analyses were performed using the R free software environment (R Development Core Team, 2023). In all statistical analyses, significance was set at a 5% risk error (Supplementary Fig. 8 and Supplementary Table 5).

All experiments performed for this manuscript were repeated at least three times. For embryo stainings, we analysed more than 50 specimens of the relevant stages (5 to 16 h of development). For cuticle preparations, more than 50 first-instar larvae were analysed. For third instar imaginal discs imaginal disc and adult testis analysis a minimum of 20 specimens were studied.

## Reporting summary
Further information on research design is available in the Nature Portfolio Reporting Summary linked to this article.

## Data availability
All data generated in this study are available within the Article and Supplementary Files. There are no data restrictions. Fly stocks can be freely obtained upon request. Source data are provided in this paper.

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

## Acknowledgements

We thank Dr. Steve Brown and Dr. Maria Luisa Rivas for helping with the isolation of the *upd0.43* cis-regulatory element, Dr. Fernando Casares and his laboratory members for help with *Episyrphus* cultures and Dr. Judith Kassis for stocks. This work was supported by Ministerio de Ciencia e Innovación grants BFU2016-76528 and PID2019-104656GB-I00 co-funded by the European Regional Development Fund (FEDER) to J.C.-G.H. and by the Spanish Ministry of Science and Innovation (PID2020-116041GB-I00) and *Leonardo Grants for Researchers and Cultural Creators 2021* from BBVA Foundation (IN[21]_CMA_BIO_0056) to I. A. The CABD is supported by a María de Maeztu Unit excellence grant CEX-2020-001088-M.

## Author contributions

S.M.-G. performed experimental studies in the embryo, S.S. performed experimental studies in the adult testis, J.M.E.-V. performed EMSA and in vitro mutagenesis, I.A. introduced the Episyrphus model. J.C.-G.H. coordinated, supervised the study and wrote the manuscript in collaboration with all authors.

## Competing interests

The authors declare no competing interests.
