## [Peer Review File · Nature Communications]

REVIEWER COMMENTS

Reviewer #1 (Remarks to the Author):

This is a very interesting manuscript on gene network co-option. Inspired by previous research on the common gene network for the formation of posterior spiracles in *Drosophila* larvae and posterior lobes in adults, they started to study the regulation of engrailed gene in the posterior spiracle formation. In fact, they found that engrailed is not involved in the development of the posterior spiracles and that the enhancer that drives its expression in the posterior spiracles is important for a function in the testis for spermatogenesis. Since there are multiple genes expressed in the testis that are expressed in the posterior spiracle and several of them share the same enhancers, it is assumed that their expression was acquired evolutionarily by gene regulatory network co-option. The research procedures are very thorough and I see no technical flaws.

Major concern

The story is very complex and I could make sense of it after careful rereading. I was able to read it carefully, especially since I have a strong interest in this field, but it may be too difficult for a wide range of readers. The word "interlocked" may be important and appropriate as a symbolic word for this study, but it is not clear what it means until after reading the whole manuscript, so it should be explained at the beginning (in the abstract).

Minor points

The Introduction begins with an individualized discussion of *Drosophila*. Because the conclusion is to present a new concept of gene network co-options, the introduction can start with a general discussion of gene network co-options, including different organisms. Otherwise, the end of the discussion makes readers feel tacked on.

At the Results, I would recommend authors include a diagram showing the regulatory relationships of the genes in each tissue. I drew my own diagram as I read. It would be easier to understand if the diagram included not only the causal regulatory relationships but also the enhancer names and whether the upstream factors have been proven to bind directly to the enhancers.

Please check if the small description of suppl fig 6B is correct. Is Lane 4 +Hop-TML, not STAT-GFP? If so, I think the authors should explain what they mean by adding Hop-TML. (STAT phosphorylated to bind DNA?)

Suppl fig 8 shows the measurement of the area of the posterior lobes, how did the authors define the basal border? Also, since it is a comparison between multiple groups, a multiple comparison test such as Tukey should be used instead of t-test.

Is there no arrow from Abd-B to en in Suppl fig 9? The text describes that the enD4.0 fragment with mutated Abd-B binding sites did not drive expression.

Reviewer #2 (Remarks to the Author):

Molina-Gil et al. present an extensive study of gene expression and regulation in a network of genes in insects that has been coopted during evolution to contribute to the formation of different body regions. The studies here are based upon previous work from others that a gene network regulating larval respiratory posterior spiracles was coopted by the male genital disc primordium, resulting in formation of a novel structure – the posterior lobe – in specific drosophilid lineages. The authors present a large body of work that includes a range of different experimental techniques, including gene expression analysis, transgenic reporter assays and DNA binding studies. The experiments were carefully done and the figures are very convincing. In particular, the expression pattern results are beautifully done and presented. The authors document a rare example of the expression of En in an anterior compartment in the

posterior region of *D. melanogaster* and identify a CRE responsible for this expression. The sections of the paper leading up and documenting this are very convincing, including the identification of trans-acting factors through a combination of bioinformatics, reporter gene expression in mutants, DNA binding assays, and mutational analysis of reporter genes. These results are presented in 2 main figures and 6 supplementary figures and form a story on their own, much of which is the focus of the general conclusions on this section in the last paragraph of the introduction are quite sound (lines 95 – 101).

The next section of the paper switches to looking at the control of *Sal* expression – although the previous DNA binding assays did not provide evidence for a direct role of *Sal* in regulating *engrailed*. The logic for looking at *Sal* regulation in this context was lost on me. The identification of a *Sal* CRE is straightforward, and the conclusion that it is *Abd-B* responsive is sound. However, there is no basis to conclude anything about the requirement during evolution for a 2-step process here. Evolutionary changes in these CREs as not even examined. Overall, this section detracts from the presentation and is confusing.

The authors then go on to generate and characterize a deletion of the *enD* CRE. This generates an unexpected finding that the posterior spiracles and genitalia are wild type-like but homozygous males are sterile. Following up on this, the authors document a role for *En* in the testis and find that some of the genes that regulate *en* in the posterior spiracle also regulate it in testis. A comparison of expression patterns of *En* and *Sal* in two other species showed that *Sal* and *En* are also expressed in the testes of *D. virilis* but not the more basally branching species *E. balteatus*. Evolutionary conclusions about the role of *En* or *Sal* made from this single comparison can only be suggestive, since many other aspects of testis morphology differ and importantly, since no functional experiments were done in any species other than *D. melanogaster*.

Overall, while the experimental results are very clear, the conclusions and “selling points” of the paper are much too speculative and are not supported by the results presented. This is clear for several aspects of the paper (as noted above), but especially for the Abstract and Discussion. Some examples are: (1) There is no evidence for “pre-adaptation” and expression pattern differences are not sufficient to make this type of conclusion. (2) Expression patterns may well vary when comparing species but attributing this to evolution of a new CRE is not justified if we don’t know how the gene is regulated in other species. Just as one possibility: maybe *D. virilis* has the same CRE but a TF is expressed differently or chromatin structure is organized differently. (3) The authors stress the significance of cooption of a GRN from one germ layer to another germ layer as being of particular significance: how so?

Minor points:

Line 55 – forward: clarify which aspects of the genetics and interactions refer to work done in *Drosophila melanogaster* (I think most of it unless specified)

Protein description too minimal: are these full length or partial proteins (eg, just the HD for *Abd-B* or the full length protein). Were they purified or are these crude extracts of the S2 transfected cells?

Reviewer #3 (Remarks to the Author):

In this manuscript, Molina-Gil and colleagues describe an intriguing case of repeated gene regulatory network co-option that involves the posterior spiracles, genital posterior lobes, and the testis. The authors investigated the unusual appearance of *engrailed* in the anterior compartment of the developing posterior spiracle and investigated the evolutionary history and molecular basis of this trait. First, they find that the *engrailed* expression seems to have evolved after the evolution of the spiracle (though only 3 species were compared, so it may be that the outgroup lineage they chose experienced a loss). They identified the enhancer of the *engrailed* locus that mediates this anterior activity and characterized its regulatory factor dependencies. Surprisingly, the deletion of this *engrailed* enhancer had no effect on the spiracle, and only a slight effect on the

genital lobe. Despite these mild effects, the authors found that flies bearing this deletion were male sterile. This led them to examine engrailed expression as well as the expression of other posterior spiracle network factors in the testes, which revealed the surprising finding that many of the same factors in this network are deployed in the testes. The authors conclude that the deployment of engrailed in the testes may have led to ectopic engrailed in the anterior A8 compartment by virtue of the interlocking nature of the co-opted gene regulatory network, leading to an expression pattern that is non-functional today, but might be a pre-adaptation for future developmental innovations.

Overall, this is a very interesting story that shows an intriguing example of gene regulatory network co-option and the surprising dispensability of a key regulatory factor's function in development. However, there are several shortcomings with the presentation of this story and experimental details that should be rectified to make it accessible to the readership of Nature Communications. Namely, I think that some of the claims should be tempered, the figures need to be more carefully presented, and missing experimental should be filled in.

Major Concerns

1. The expression of engrailed in the anterior compartment seems to be unique to the Drosophilids tested, but the authors did not establish that enhancer mediating this pattern is necessarily novel. To do so, one would want to see how the orthologous region from species such as *E. balteatus* behaves in a reporter assay. Clearly this is not a required experiment on the timeline of revision, but claims about the evolutionary origins of the enD enhancer should be moderated.
2. The authors call the anterior expression of engrailed a "novelty", which may conflict with the common conceptions of what a novelty is (i.e. many researchers think of novelty as new physical traits, often those facilitating adaptation or radiation and not just expression patterns). I think the authors should either call this an "expression novelty" (or some other term) that clearly differentiates or better define what their definition of a novelty.
3. The authors suggest that the co-option of the spiracle network had adaptive value in the testes (I am assuming because flies lacking the enD enhancer are sterile) – for example – L33, L389. However, this might not be true and is actually likely wrong. I imagine that insects had perfectly good ways to produce fertile sperm before the posterior spiracle network was co-opted to testes development. It seems far more likely that drift and gene regulatory network turnover caused the expression of this network to take over the roles of other factors to maintain this critical function.
4. The authors should acknowledge that because their trait is an expression trait, which was only examined in three species, that the evolutionary scenario described might actually be a loss in *E. balteatus*. This is not so problematic for phenotypes that can be observed non-molecularly, as many more species can be examined. The authors note differences in testis morphology that might have resulted from network co-option but provide no evidence that this is the case.

Minor Concerns

It is nearly impossible to interpret the results presented in Figure 5 without better labels of the insets etc. What is EnDds-GFP? Multiple black and white insets are shown without any indication of what the stains are.

The interlocking network concept is not clearly explained – please define it more clearly in the manuscript. Perhaps a model figure explaining the idea would help the reader.

The addition of sal enhancers to the paper seems somewhat tangential to the main point of the paper and it is worth considering relegating to the supplement to maintain readability.

There is some background on gametogenesis and testis development that could be explained better for the audience. The paper spans a multitude of organ systems and would benefit readers from diverse experimental backgrounds. I would recommend putting the Figure 5A schematic into Figure 4. I think there is room in that drawing to label the different stages rather than use numbers. This would greatly increase readability.

Overall, many of the figure panels lack labels, and this detracts from readability. Describing what is in each panel in the legend makes digesting the manuscript much harder for most readers.

Figure 1G – it is very hard to see the extent of the spalt stain in *E. balteatus*. This is a very important result, as it establishes that difference in anterior compartment engrailed. Anything to make this panel bigger or easier to see would help. A cartoon that draws out what we are looking at would be very nice.

Figure 3G-H should show a control embryo at the same stage for comparison. Please list in the legend what driver was used for mis-expression of Abd-B isoforms.

Figure 4 was very hard to understand at first glance. It would help to label on the figure what each thing is. As there is quite a bit of dark space in the figure, there appears to be room. Alternately a well labeled cartoon would help the reader understand what they are looking at. The figure itself looks hastily assembled with panels not lining up harmoniously.

Figure 6 is also very hard to interpret. Showing the separate channels is critical for the reader to assess the differences in presence/absence of posterior spiracle network genes. Panels could be much more nicely assembled in this figure.

Sup Fig3 – for the dissection of enD, it would really help to show close-up images of the spiracle to see the differences in the subdivisions of the enD region (akin to the closeups in Fig 1).

For mutation of binding sites (Lines 196-203), please state in the manuscript what the nucleotide alterations to the site are. This is way more useful to interpreting these experiments than simply presenting that information in a table of primers.

For descriptions of the sperm defects (lines 252-258), more context is needed for the reader to understand the nature of these defects. Just a sentence or two introducing the process of sperm development would go a long way. A schematic of the process in figure 4 would be even better.

Sup Figure 6 is hard to interpret – it is not clear what is present in every lane of the gel, and the basis of this experiment, especially the STAT EMSA is not explained. What are all the components added to lane 4 of panel B? Is the amnioserosa activity of enD an endogenous function of engrailed?

Sup Figure 8 – no units are presented in graph or table. Are these pixels? Nanometers? I would suggest that showing the datapoints on the graph would make the figure simpler and convey the same information more easily.

Supplementary Figure 9 – show separate channels rather than merges as it is very difficult to see the expression patterns that are presented.

The enX enhancer is suggested to represent “sensory elements”, but the images presented in Sup Fig 2 don’t support this claim so clearly. Perhaps showing a counterstain or zoomed in image would help?

Lines 152-154 “As all reporters containing this regulatory element drive similar expression....we employ them indistinctly to study en regulation...” What do you mean? Does this mean that enD in different figures refers to different versions of the construct? If so, I think it would help to make those references more explicit throughout figures, text and legend.

RESPONSE TO REVIEWERS' COMMENTS

Reviewer #1 (Remarks to the Author):

Comment: This is a very interesting manuscript on gene network co-option. Inspired by previous research on the common gene network for the formation of posterior spiracles in *Drosophila* larvae and posterior lobes in adults, they started to study the regulation of engrailed gene in the posterior spiracle formation. In fact, they found that engrailed is not involved in the development of the posterior spiracles and that the enhancer that drives its expression in the posterior spiracles is important for a function in the testis for spermatogenesis. Since there are multiple genes expressed in the testis that are expressed in the posterior spiracle and several of them share the same enhancers, it is assumed that their expression was acquired evolutionarily by gene regulatory network co-option. The research procedures are very thorough and I see no technical flaws.

Major concern

The story is very complex and I could make sense of it after careful rereading. I was able to read it carefully, especially since I have a strong interest in this field, but it may be too difficult for a wide range of readers. The word "interlocked" may be important and appropriate as a symbolic word for this study, but it is not clear what it means until after reading the whole manuscript, so it should be explained at the beginning (in the abstract).

Response: We could not add an explanation in the abstract as due to the Nature Communications's formatting requirements we had to cut its length from 200 words down to 150. To comply with the reviewer's suggestion, we have added an explanation of the interlocking concept in the last paragraph of the introduction (lines 111-120). We have also added a new scheme (Fig. 7b) requested by Reviewer 3 that facilitates understanding the interlocking of co-opted gene networks.

Minor points

Comment: The Introduction begins with an individualized discussion of *Drosophila*. Because the conclusion is to present a new concept of gene network co-options, the introduction can start with a general discussion of gene network co-options, including different organisms. Otherwise, the end of the discussion makes readers feel tacked on.

Response: We have added at the beginning of the introduction a more general paragraph on co-option in other organisms (lines 44-54).

At the Results, I would recommend authors include a diagram showing the regulatory relationships of the genes in each tissue. I drew my own diagram as I read. It would be easier to understand if the diagram included not only the causal regulatory relationships but also the enhancer names and whether the upstream factors have been proven to bind directly to the enhancers.

Response: The new scheme presented as Fig. 7b includes this diagram, although we did not show the enhancer names and direct binding data as it would make it too complex.

Please check if the small description of suppl fig 6B is correct. Is Lane 4 +Hop-TML, not

STAT-GFP? If so, I think the authors should explain what they mean by adding Hop-TML. (STAT phosphorylated to bind DNA?)

Response: We apologize for the confusing and incomplete description. We have labelled the EMSA panels more clearly adding a key over each lane describing the proteins expressed in the different cell extracts tested. In the figure legend we now say that Hop TML is an activated JAK kinase used to phosphorylate STAT into its active form.

Suppl fig 8 shows the measurement of the area of the posterior lobes, how did the authors define the basal border? Also, since it is a comparison between multiple groups, a multiple comparison test such as Tukey should be used instead of t-test.

Response: We have repeated the posterior lobe quantification to do the measurements as described by Frazee and Masly 2015 Ecol Evol. 2015 Sep 23;5(19):4437-50. doi: 10.1002/ece3.1721. eCollection 2015 Oct. The outline of each posterior lobe was traced manually using Fiji ImageJ and enclosed with an artificial baseline drawn in line with the lateral plate. With the new data we have performed an analysis of variance and Tukey's post hoc test as suggested by the reviewer. Due to the new quantification criteria the image looks slightly different but it does not change the result. As suggested by reviewer 3, we have removed the data from the figure and included them as single points in the box plot. The raw data can be found as Supplementary table 4.

Is there no arrow from Abd-B to en in Suppl fig 9? The text describes that the enD4.0 fragment with mutated Abd-B binding sites did not drive expression.

Response: While revising this manuscript, we found a mistake where a GFP expressing reporter gene shown in Fig.5g (the enD 0.4 construct with the putative Abd-B binding sites mutated) had been mistakenly stained with anti-mCherry. After repeating the staining, we have found that the construct is expressed in the HCC and have therefore changed the panel and corrected the figure legend and corresponding text in Results and Discussion sections where we now say:

Line 313-320:

"To test the extent to which the enD enhancer regulation is conserved, we analysed the expression of the Abd-B and STAT mutated binding site reporters. As it could be expected from the absence of Abd-B expression in the HCC, mutating the Abd-B binding sites does not affect the enhancer testis expression (Fig. 5g). Mutation of the STAT-binding sites does not affect the construct's expression either (Fig. 5h), this may be because it still has a secondary input from the gene network as it happens in the posterior spiracles (see above). ~~while mutating the Abd-B binding sites abolishes enD0.4 expression in HCC (Fig. 5g, and discussion).~~"

And have deleted in the Discussion section the following sentence from lines 356-358:

~~"In this scenario, the observed requirement of the Abd-B binding sites for enD expression in the HCC could be due to the site overlapping that of another transcription factor active in the HCCs."~~

As a result, there is no arrow from Abd-B to en in Sup. Fig. 9 as this figure only shows testis interactions. The arrow, however, is depicted in the new panel Fig.7b in the posterior spiracle column.

Reviewer #2 (Remarks to the Author):

Molina-Gil et al. present an extensive study of gene expression and regulation in a network of genes in insects that has been coopted during evolution to contribute to the formation of different body regions. The studies here are based upon previous work from others that a gene network regulating larval respiratory posterior spiracles was coopted by the male genital disc primordium, resulting in formation of a novel structure – the posterior lobe – in specific drosophilid lineages.

The authors present a large body of work that includes a range of different experimental techniques, including gene expression analysis, transgenic reporter assays and DNA binding studies. The experiments were carefully done and the figures are very convincing. In particular, the expression pattern results are beautifully done and presented.

The authors document a rare example of the expression of En in an anterior compartment in the posterior region of *D. melanogaster* and identify a CRE responsible for this expression. The sections of the paper leading up and documenting this are very convincing, including the identification of trans-acting factors through a combination of bioinformatics, reporter gene expression in mutants, DNA binding assays, and mutational analysis of reporter genes. These results are presented in 2 main figures and 6 supplementary figures and form a story on their own, much of which is the focus of the general conclusions on this section in the last paragraph of the introduction are quite sound (lines 95 – 101).

The next section of the paper switches to looking at the control of Sal expression – although the previous DNA binding assays did not provide evidence for a direct role of Sal in regulating engrailed. The logic for looking at Sal regulation in this context was lost on me. The identification of a Sal CRE is straightforward, and the conclusion that it is Abd-B responsive is sound. However, there is no basis to conclude anything about the requirement during evolution for a 2-step process here. Evolutionary changes in these CREs as not even examined. Overall, this section detracts from the presentation and is confusing.

Response: We agree that the Sal expression section slows down the flow of the manuscript, however we want to keep it in the Results section because it describes *sal2.1*, a *spalt* enhancer functional both in the posterior spiracles and in the testis head cyst cells. Therefore *sal2.1* regulation is important as shows that not only the same genes are expressed in the two organs, but that they are activated using the same enhancers, reinforcing the argument that the gene network has been co-opted.

Regarding the proposal of a 2-step process in evolution, this is not based on the enhancer, but on the fact that Sal is expressed both in *Episyrphus* and in *Drosophila*, but Engrailed only in *Drosophila*, suggesting that the activation of both genes occurred at two different evolutionary stages. Although Reviewer 3 indicates that the absence of En in *Episyrphus* could be due to its secondary loss, published data of En expression in other Diptera species have shown that En expression is restricted to a posterior segmental stripe (Clearly observed in *Bactrocera dorsalis* and in *Lucila sericata* (*Int. J. Dev. Biol.* **61**: 439-450 (2017) doi: 10.1387/ijdb.160277sb; *Developmental Dynamics* 235:347–360, 2006). We indicated this in the previous manuscript and also in the revised version in lines 384-387.

The authors then go on to generate and characterize a deletion of the enD CRE. This generates an unexpected finding that the posterior spiracles and genitalia are wild type-like but homozygous males are sterile. Following up on this, the authors document a role for En in the testis and find that some of the genes that regulate en in the posterior spiracle also regulate it in testis. A comparison of expression patterns of En and Sal in two other species showed that Sal and En are also expressed in the testes of *D. virilis* but not the more basally branching species *E. balteatus*. Evolutionary conclusions about the role of En or Sal made from this single comparison can only be suggestive, since many other aspects of testis

morphology differ and importantly, since no functional experiments were done in any species other than *D. melanogaster*.

Overall, while the experimental results are very clear, the conclusions and “selling points” of the paper are much too speculative and are not supported by the results presented. This is clear for several aspects of the paper (as noted above), but especially for the Abstract and Discussion. Some examples are:

(1) There is no evidence for “pre-adaptation” and expression pattern differences are not sufficient to make this type of conclusion.

Response: The reviewer must note that in this manuscript we use the concept of pre-adaptation *sensu stricto*, as opposed to the associated term of *exaptation*. While in *exaptation*, the co-opted character had a previous selective function (as is the case of feathers that initially served for thermal isolation before they were co-opted for flight), in this case the expression of *Engrailed* in the posterior spiracle has no function. Thus, we propose that *engrailed* gene expression in the anterior compartment of the posterior spiracles is a pre-adaptive *expression novelty* resulting from the co-opted gene networks’ being interlocked. We consider this to be a true pre-adaptation as in the posterior spiracle anterior cells the *Engrailed* gene provides no selective function but may acquire it in the future by recruiting downstream targets. The evidence we present for pre-adaptation in the posterior spiracles comes from the finding that a rather sophisticated regulatory gene network is activating very precisely in time and space the expression of an important developmental transcription factor in the anterior part of the A8 segment where it has no function whatsoever for its development. The fact that this happens would be paradoxical had not been by the discovery of the gene network interlocking.

We have expanded this explanation at the end of the discussion lines 432-439 where we say:

“Thus, we propose that co-option can lead to sensu stricto pre-adaptation cases, as opposed to the associated term of exaptation. While in exaptation the co-opted character had a previous selective function that has been recruited to perform a novel function (i.e. heat shock proteins recruited to form the eye crystallin, or feathers that could have served for heat regulation or sexual display before being co-opted for flight) cases like the expression of Engrailed in the posterior spiracle provide no selective advantage but could conceivably acquire it in the future as it has happened in the anterior wing of several Diptera where En has acquired a new role in wing pigmentation.”

(2) Expression patterns may well vary when comparing species but attributing this to evolution of a new CRE is not justified if we don’t know how the gene is regulated in other species. Just as one possibility: maybe *D. virilis* has the same CRE but a TF is expressed differently or chromatin structure is organized differently.

Response: The reviewer is correct. With our results we cannot tell if the *enD* enhancer exists as a silent cryptic enhancer in species like *Episyrphus* and has become active in *Drosophila* due to the recruitment to the spiracle of a transcription factor or because of changes in chromatin accessibility. To accommodate for this, we have modified in Discussion lines 391-396:

“Taken together, Engrailed expression in the anterior compartment of A8 evolved after functional posterior spiracles had already formed in dipteran larvae and its origin had to do with testis evolution. Future experiments should establish if this occurred due to the de novo appearance of the enD CRE, or to a silent enD cryptic enhancer becoming activated due to the

recruitment of a new transregulatory factor in the spiracles/testis gene network or due to changes in chromatin accessibility.”

(3) The authors stress the significance of cooption of a GRN from one germ layer to another germ layer as being of particular significance: how so?

Response: Previous characterized cases of gene network co-option occur among tissues formed by the same germ layer. Although we believe we present the first case of gene network co-option between different germ layers, we agree with the reviewer that given the small number of existing examples, this may turn out to be rather normal. Therefore, in the abstract and at the end of the introduction we have toned down the stress on the gene network having been co-opted to a different germ layer. In lines 108-110 we now say: “*Our work presents an example of repeated sequential gene network co-option involving tissues of different germ layers, (...)*”

And in discussion (lines 339-342) we have modified the following paragraph:

“Previous work showed that the ectodermally expressed posterior spiracle gene network had been co-opted to the ectodermal male genitalia in the Drosophila melanogaster clade. Here we show the same gene network has also been co-opted to the mesodermal head cyst cells in Drosophila”.

Minor points:

Line 55 – forward: clarify which aspects of the genetics and interactions refer to work done in *Drosophila melanogaster* (I think most of it unless specified)

Response: We have specified that all genetic interactions have been analyzed in *D. melanogaster*. (Lines 106)

Protein description too minimal: are these full length or partial proteins (eg, just the HD for Abd-B or the full length protein). Were they purified or are these crude extracts of the S2 transfected cells?

Response: We have clarified in the Figure legend (line 1095), in Results (line 204) and in the Materials and methods section (lines 500-504) that we used S2 cell crude extract and that all transfected proteins were full-length.

Reviewer #3 (Remarks to the Author):

In this manuscript, Molina-Gil and colleagues describe an intriguing case of repeated gene regulatory network co-option that involves the posterior spiracles, genital posterior lobes, and the testis. The authors investigated the unusual appearance of engrailed in the anterior compartment of the developing posterior spiracle and investigated the evolutionary history and molecular basis of this trait. First, they find that the engrailed expression seems to have evolved after the evolution of the spiracle (though only 3 species were compared, so it may be that the outgroup lineage they chose experienced a loss). They identified the enhancer of the engrailed locus that mediates this anterior activity and characterized its regulatory factor dependencies. Surprisingly, the deletion of this engrailed enhancer had no effect on the spiracle, and only a slight effect on the genital lobe. Despite these mild effects, the authors found that flies bearing this deletion were male sterile. This led them to examine engrailed

expression as well as the expression of other posterior spiracle network factors in the testes, which revealed the surprising finding that many of the same factors in this network are deployed in the testes. The authors conclude that the deployment of engrailed in the testes may have led to ectopic engrailed in the anterior A8 compartment by virtue of the interlocking nature of the co-opted gene regulatory network, leading to an expression pattern that is non-functional today, but might be a pre-adaptation for future developmental innovations.

Overall, this is a very interesting story that shows an intriguing example of gene regulatory network co-option and the surprising dispensability of a key regulatory factor's function in development. However, there are several shortcomings with the presentation of this story and experimental details that should be rectified to make it accessible to the readership of Nature Communications. Namely, I think that some of the claims should be tempered, the figures need to be more carefully presented, and missing experimental should be filled in.

Major Concerns

1. The expression of engrailed in the anterior compartment seems to be unique to the Drosophilids tested, but the authors did not establish that enhancer mediating this pattern is necessarily novel. To do so, one would want to see how the orthologous region from species such as *E. balteatus* behaves in a reporter assay. Clearly this is not a required experiment on the timeline of revision, but claims about the evolutionary origins of the enD enhancer should be moderated.

Response: This comment agrees with Reviewer 2. With our results we cannot tell if the *enD* enhancer exists as a silent cryptic enhancer in species like *Episyrphus* that has become active due to the recruitment to the spiracle of a transcription factor or because the chromatin became accessible. To accommodate for this, we have modified the Discussion (lines 391-396):

“Taken together, Engrailed expression in the anterior compartment of A8 was acquired after functional posterior spiracles already existed in dipteran larvae and its origin had to do with testis evolution. Future experiments should establish if this occurred due to the de novo appearance of the enD CRE, or to a silent enD cryptic enhancer becoming activated due to the recruitment of a new transregulatory factor in the spiracles/testis gene network or to changes in chromatin accessibility.”

2. The authors call the anterior expression of engrailed a “novelty”, which may conflict with the common conceptions of what a novelty is (i.e. many researchers think of novelty as new physical traits, often those facilitating adaptation or radiation and not just expression patterns). I think the authors should either call this an “expression novelty” (or some other term) that clearly differentiates or better define what their definition of a novelty.

Response: In abstract and in lines 32, 111 and 132 we have now written “expression novelty” to clarify that we do not refer to a physical trait.

3. The authors suggest that the co-option of the spiracle network had adaptive value in the testes (I am assuming because flies lacking the enD enhancer are sterile) – for example – L33, L389. However, this might not be true and is actually likely wrong. I imagine that insects had perfectly good ways to produce fertile sperm before the posterior spiracle network was co-opted to testes development. It seems far more likely that drift and gene regulatory

network turnover caused the expression of this network to take over the roles of other factors to maintain this critical function

Response: We only partially agree with this comment.

We agree that finding out how a functional *enD* enhancer driving Engrailed expression in the testis appeared would be of great interest. The reviewer is right to say that with our data we cannot say if the spiracle gene network recruitment to the testis was due to selective advantage, drift or gene regulatory turn over. Therefore, we have taken out the comment about selective advantage from the abstract.

Despite this, we do not agree with the reviewer's asseveration that we are probably wrong. There is no doubt that Diptera produced fertile sperms before the posterior spiracle gene network was co-opted to the testes. However, major changes in sperm size and gonad shape as those happening after the divergence of *Episyrphus* and *Drosophila* may have required not only genetic drift but also the selection of testis variants to maintain fertility. This probably was achieved by several means, among which the co-option of the posterior spiracle gene network and the selection of various genes may have had adaptive value.

To explain this, we have added to the discussion the following sentence (line 410-414):

“The co-option of the posterior spiracle cascade to the testis and the recruitment of Engrailed expression may have been the result of the major sperm size and gonad morphology changes occurring at some point after the divergence of Episyrphus and Drosophila that may have required the selection of new genetic variants to maintain fertility.”

4. The authors should acknowledge that because their trait is an expression trait, which was only examined in three species, that the evolutionary scenario described might actually be a loss in *E. balteatus*. This is not so problematic for phenotypes that can be observed non-molecularly, as many more species can be examined. The authors note differences in testis morphology that might have resulted from network co-option but provide no evidence that this is the case.

Response: The expression of engrailed has been analyzed in other species like *Bactrocera dorsalis* and *Lucila sericata* which show no anterior expression in A8 (**Int. J. Dev. Biol. 61:** 439-450 (2017) doi: 10.1387/ijdb.160277sb; *Developmental Dynamics* 235:347–360, 2006) suggesting that what occurred was not a loss of expression in *Episyrphus* but gain of expression in *Drosophila*.

We report this in the manuscript (lines 384-387):

“En activation in A8a associated to the posterior spiracle is present in D. virilis, but not in E. balteatus, suggesting it originated in the higher Diptera (Brachicera). This is supported by analyses showing that in other Diptera like Bactrocera dorsalis (Tephritidae) and Lucilia sericata (Calliphoridae) that present well-developed larval posterior spiracles engrailed expression is restricted to the posterior A8 segment.”

We agree that identifying what are the functions of each element of the co-opted network is of great interest and this is the topic of our future research.

Minor Concerns

It is nearly impossible to interpret the results presented in Figure 5 without better labels of the insets etc. What is EnDds-GFP? Multiple black and white insets are shown without any indication of what the stains are.

Response: We have labelled the insets in Fig. 5 and we have also included a new Supplementary table 3 listing all the reporter constructs used in this manuscript.

enD-ds is one of the enhancer variants made in this study to activate the GFP reporter. We have homogenized the labels used in all figures. We have clarified in the text that the *enD*, *enD-ds* and *enD-0.4* enhancers were used to express lac-Z, GFP or mCherry respectively and that we used the three reporter lines indistinctly as they drive the same spatio-temporal expression. In line 165-168 we say:

“As all reporters containing these regulatory elements drive similar expression (Supplementary Fig. 3), in what follows we employ them indistinctly to study en regulation in the posterior spiracle using either enD-lacZ, enDds-GFP or enD0.4-mCherry as specified in each particular figure.”

In Supplementary Fig. 3 we have also added two new panels (f-g) presenting embryos carrying combinations of constructs double stained to show the expression of all transgenic enD variant lines is comparable.

The interlocking network concept is not clearly explained – please define it more clearly in the manuscript. Perhaps a model figure explaining the idea would help the reader.

Response: This comment agrees with Reviewer 1. We now explain at the end of the introduction section (lines 111-120) the concept of gene network interlocking. We have also included a model in Fig 7b explaining the co-option of the posterior spiracle gene network to the testis HCC and the appearance of *en* expression in both organs caused by the gene network interlocking.

The addition sal enhancers to the paper seems somewhat tangential to the main point of the paper and it is worth considering relegating to the supplement to maintain readability.

Response: As explained before, we believe the *sal2.1* enhancer is important to demonstrate that the gene network has been co-opted. Therefore, we prefer this section to remain where it is even if it makes reading less agile.

There is some background on gametogenesis and testis development that could be explained better for the audience. The paper spans a multitude of organ systems and would benefit readers from diverse experimental backgrounds. I would recommend putting the Figure 5A schematic into Figure 4. I think there is room in that drawing to label the different stages rather than use numbers. This would greatly increase readability.

Response: We have moved the testis scheme from Fig. 5A to Fig. 4 and, as suggested by the reviewer, have also moved the explanation about testis development that previously was in the discussion section (now deleted lines 342-348) to the beginning of the result section titled “*Posterior spiracle gene network expression in the testis*” (lines 280-291), which should help readers from other experimental backgrounds. It now reads:

“In Drosophila, the germ cell niche is located at the testis apex in a structure known as the hub (Fig. 4a). When the germ stem cell divides, it gives rise to a sperm progenitor cell (gonialblast) that separates from the hub and becomes encapsulated by two highly specialized mesodermal cells (the cyst cells). The cyst cells do not proliferate but will protect and signal to the gonialblast as it divides^{35,36}. Inside each cyst, the gonialblast experiences four mitoses and a meiosis generating 64 clonal of spermatids that elongate (Fig. 4a). When the spermatids individualize to give rise to the spermatozoa, the two cyst cells differentiate to form a head cyst cell (HCC) that forms an Actin basket holding tightly the 64 sperm heads, and a tail cyst cell (TCC) that elongates to surround the growing sperm tails (Fig. 4b). After the 64 spermatozoa individualise, they coil at the testis terminal region where, during the phase known as

spermiation, generate forces that lead to their liberation from the cyst and exit to the seminal vesicle (Fig. 4c).^{37,38}”

Overall, many of the figure panels lack labels, and this detracts from readability. Describing what is in each panel in the legend makes digesting the manuscript much harder for most readers.

Response: We have added labels to the figures.

Figure 1G – it is very hard to see the extent of the spalt stain in *E. balteatus*. This is a very important result, as it establishes that difference in anterior compartment engrailed. Anything to make this panel bigger or easier to see would help. A cartoon that draws out what we are looking at would be very nice.

Response: To facilitate the comparison between the three species, we have added in panels 1e'-1f' an arrow pointing at the Sal stigmatophore expression and an asterisk to mark the position of the posterior spiracle's opening.

Figure 3G-H should show a control embryo at the same stage for comparison. Please list in the legend what driver was used for mis-expression of Abd-B isoforms.

Response: We have included a control embryo as panel f. The scheme that was previously included in this figure as panel F has been integrated in Fig. 1b. We have indicated in the legend that the line used for ectopic expression is *69B-Gal4*.

Figure 4 was very hard to understand at first glance. It would help to label on the figure what each thing is. As there is quite a bit of dark space in the figure, there appears to be room. Alternately a well labeled cartoon would help the reader understand what they are looking at. The figure itself looks hastily assembled with panels not lining up harmoniously.

Response: We have modified Fig. 4 to make it simpler to understand, adding the cartoon that was previously presented in Fig. 5A as panel 4b. We have further modified this figure showing a close up of the wild type testes that were in the original figure, but focusing only on one of the two testes where we have labelled the different structures. We have removed what used to be panel B (as this information is well conveyed by the close-up shown in panel d) to make room for the terminal testis cartoon. In panels c and d we now show a close-up of the control and mutant testis terminal region. Panels 4e and f have been maintained as in the original submitted version. All panels have been flipped horizontally to have posterior to the right. The figure panels have been carefully aligned and cropped to obtain a more harmonious organization.

Figure 6 is also very hard to interpret. Showing the separate channels is critical for the reader to assess the differences in presence/absence of posterior spiracle network genes. Panels could be much more nicely assembled in this figure.

Response: The figure panels have been aligned and cropped to obtain a more harmonious organization. The separate channels of the boxed region are shown as insets with the single channel images labelled.

Sup Fig3 – for the dissection of enD, it would really help to show close-up images of the

spiracle to see the differences in the subdivisions of the enD region (akin to the closeups in Fig 1).

Response: The expression of all three constructs is almost identical. To illustrate this, we have added two panels to Sup Fig.3 showing posterior spiracle close ups as those requested by the reviewer showing a double staining of *enD-lacZ* with *enDds-GFP* (panel f), and of *enD-lacZ* with *enD0.4-mCherry* (panel g).

For mutation of binding sites (Lines 196-203), please state in the manuscript what the nucleotide alterations to the site are. This is way more useful to interpreting these experiments than simply presenting that information in a table of primers.

Response: We have described the STAT (lines 211-212) and Abd-B binding site (lines 216-217) mutations in the Results and Material and methods sections (line 533-547)

For descriptions of the sperm defects (lines 252-258), more context is needed for the reader to understand the nature of these defects. Just a sentence or two introducing the process of sperm development would go a long way. A schematic of the process in figure 4 would be even better.

Response: We have moved the testis terminal region scheme from Fig5A to Fig4b. We have also labelled more clearly the control testis shown in Fig.4a and added a description of spermatogenesis in the text referring to panels 4a-c, which describe the relevant stages of spermatogenesis more clearly (lines 280-291).

Sup Figure 6 is hard to interpret – it is not clear what is present in every lane of the gel, and the basis of this experiment, especially the STAT EMSA is not explained. What are all the components added to lane 4 of panel B? Is the amnioserosa activity of enD an endogenous function of engrailed?

Response: We apologize for the confusing and incomplete description. We have labelled the EMSA panels more clearly adding on top of each lane a key describing the proteins expressed in the different cell extracts used for each retardation experiment. In the figure legend we now say that Hop TML is an activated JAK kinase used to phosphorylate STAT into its active DNA binding form. In the Supplementary Figure 3 and 5 legends we also clarify the expression of enD derivatives in the amnioserosa is not an En endogenous characteristic (line 1103-1104 and line 1062-1063).

Sup Figure 8 – no units are presented in graph or table. Are these pixels? Nanometers? I would suggest that showing the datapoints on the graph would make the figure simpler and convey the same information more easily.

Response: We now specify the data are in pixels. We have included the datapoints on the graph as suggested. We have reanalyzed the data using a Tukey test as suggested by Reviewer 1. We have removed the raw data to a Supplementary Table 4 and shown the data as points in the graph.

Supplementary Figure 9 – show separate channels rather than merges as it is very difficult to see the expression patterns that are presented.

Response: We have included close ups of cells showing separate channels as insets.

The enX enhancer is suggested to represent “sensory elements”, but the images presented in Sup Fig 2 don’t support this claim so clearly. Perhaps showing a counterstain or zoomed in image would help?

Response: We believe these cells to be spiracle sensory elements due to their position and shape. We have clarified this in the figure legend (line 1045). However, as this construct is not analyzed in the manuscript, we prefer leaving the figure as it is rather than adding a panel showing a double staining for neural cell marker which is peripheral to the manuscript.

Lines 152-154 “As all reporters containing this regulatory element drive similar expression....we employ them indistinctly to study en regulation...” What do you mean? Does this mean that enD in different figures refers to different versions of the construct? If so, I think it would help to make those references more explicit throughout figures, text and legend.

Response: We have explained this more clearly (lines 165-168), where we say:
“As all reporters containing these regulatory elements drive similar expression (Supplementary Fig. 3f-g), in what follows we employ them indistinctly to study en regulation in the posterior spiracle using either enD-lacZ, enD-ds-GFP or enD-0.4-mCherry as specified in each particular figure.”

We have also added two panels to Sup Fig.3 showing posterior spiracle close ups presenting a double staining of enD-lacZ with enDds-GFP (panel f), and of enD-lacZ with enD0.4-mCherry (panel g) that illustrate the almost identical expression driven by these transgenic lines.

DELETED REFERENCES: We have deleted the following references to comply with the 70-reference maximum requirement. Deleting these references is not affecting the manuscript, as similar information is provided by the remaining 70 references.

- 7 — Simoes, S. *et al.* Compartmentalisation of Rho regulators directs cell invagination during tissue morphogenesis. *Development* **133**, 4257–4267, doi:10.1242/dev.02588 (2006).
- 16 — Clark, E. & Akam, M. Odd-paired controls frequency doubling in *Drosophila* segmentation by altering the pair-rule gene regulatory network. *Elife* **5**, doi:10.7554/eLife.18215 (2016).
- 25 — Sandelin, A., Alkema, W., Engstrom, P., Wasserman, W. W. & Lenhard, B. JASPAR: an open access database for eukaryotic transcription factor binding profiles. *Nucleic Acids Res* **32**, D91–94, doi:10.1093/nar/gkh012 (2004).
- 28 — Kuhnlein, R. P., Bronner, G., Taubert, H. & Schuh, R. Regulation of *Drosophila* spalt gene expression. *Mech Dev* **66**, 107–118, doi:10.1016/s0925-4773(97)00103-2 (1997).
- 31 — Zavortink, M. & Sakonju, S. The morphogenetic and regulatory functions of the *Drosophila* Abdominal-B gene are encoded in overlapping RNAs transcribed from separate promoters. *Genes Dev* **3**, 1969–1981, doi:10.1101/gad.3.12a.1969 (1989).

- 33—Foronda, D., Estrada, B., de Navas, L. & Sanchez-Herrero, E. Requirement of Abdominal-A and Abdominal-B in the developing genitalia of *Drosophila* breaks the posterior downregulation rule. *Development* **133**, 117–127, doi:10.1242/dev.02173 (2006).
- 52—Keys, D. N. *et al.* Recruitment of a hedgehog regulatory circuit in butterfly eyespot evolution. *Science* **283**, 532–534, doi:10.1126/science.283.5401.532 (1999).
- 62—Rogers, B. T. & Kaufman, T. C. Structure of the insect head as revealed by the EN protein pattern in developing embryos. *Development* **122**, 3419–3432, doi:10.1242/dev.122.11.3419 (1996).
- 80—Sotillos, S., Espinosa-Vazquez, J. M., Foglia, F., Hu, N. & Hombria, J. C. An efficient approach to isolate STAT regulated enhancers uncovers STAT92E fundamental role in *Drosophila* tracheal development. *Dev Biol* **340**, 571–582, doi:10.1016/j.ydbio.2010.02.015 (2010).
- 81—Fisher, C. L. & Pei, G. K. Modification of a PCR-based site-directed mutagenesis method. *Biotechniques* **23**, 570–571, 574, doi:10.2144/97234bm01 (1997).
- 82—Forloni, M., Liu, A. Y. & Wajapeyee, N. Megaprimer Polymerase Chain Reaction (PCR)-Based Mutagenesis. *Cold Spring Harb Protoc* **2019**, doi:10.1101/pdb.prot097824 (2019).
- 83—Sanchez-Higueras, C., Sotillos, S. & Castelli-Gair Hombria, J. Common origin of insect trachea and endocrine organs from a segmentally repeated precursor. *Curr Biol* **24**, 76–81, doi:10.1016/j.cub.2013.11.010 (2014).

REVIEWERS' COMMENTS

Reviewer #2 (Remarks to the Author):

I appreciate all of the changes made and the responses to the reviewers' critiques. I feel the manuscript is now essentially acceptable for publication. I have some minor suggestions, mainly on language, below:

Line 167 "employ them indistinctly" Wording does not make sense

Line 179 "as well as" should be "or"

Line 192 - 194: that result could also suggest the presence of a repressor (no need to change the text)

Line 201: add "that" in observed "that" activated

Line 256: "To find out enD requirement during development" Poor wording

"induced" should be "generated"

I would also add "in the endogenous gene" to distinguish that from a reporter transgene mutation, which is really different

Line 280: add melanogaster

Line 315: remove "it" from "As it could"

Line 317: Start new sentence with "This may be"

Line 318: remove "it" before as

Line 326: change "sperms" to "sperm"

Line 358 - 360: I do not know of Abd-B ever being implicated in epigenetic regulation. Is this a pure speculation or is it based on a known activity of Abd-B?

Line 438: remove "it" after as

Reviewer #3 (Remarks to the Author):

The manuscript is greatly improved in response to reviewer feedback. Readability of the abstract is still a major issue and I would encourage the authors to think more carefully about the following:

1. Lines 32 - 34 of abstract

"We show that a *Drosophila* expression novelty, the activation of an engrailed cis-regulatory element for sperm liberation, has influenced its organ of origin due to gene network sharing. This network "interlocking" resulted in the posterior compartment determining factor Engrailed expressed in the anterior compartment of the 36 A8 segment without any functional consequence."

Are not very clear to understand for someone reading the abstract. Introducing the term "interlocking" in the abstract in quotes without defining it will disinclude less-informed readers. The other problem is that the expression novelty is not defined - are you defining the activation of en in the anterior compartment of the spiracle an expression novelty? Or is it the function that is required for sperm liberation? Clearly it is both, but the abstract does not specify with any clarity, while instead trying to drive high level concepts the reader likely won't understand. I would also strongly consider a more explicit statement on what GRN co-option is, as the first sentence of the

abstract requires an understanding that most readers will not be familiar with. It is necessary to spend two sentences setting up the contrast between single gene co-options and whole network co-options? In this reviewer's mind, I find the dichotomy unhelpful if none of the terms being discussed are defined sufficiently in the abstract. I'll note that the summary presented at the end of the introduction (lines 102-118) is much more readable, and could be used as a guide to condense into a more readable abstract.

2. Ln 61 typo – ananassae

3. Ln 326-328 I still think it unlikely that expression of engrailed itself caused differences in testis/sperm morphology. Is the morphology of the enD deletion testis more like that of *Episyrphus balteatus*? If so, it is not noted in the paper. It strikes me as a fanciful point to end the results section on.

4. The final paragraph of the discussion (lines 397-440) is very long! I would recommend splitting the two ideas it contains. Maybe break to a new paragraph at line 414, or maybe line 425.

RESPONSE TO REVIEWERS' COMMENTS

Reviewer #2 (Remarks to the Author):

I appreciate all of the changes made and the responses to the reviewers' critiques. I feel the manuscript is now essentially acceptable for publication. I have some minor suggestions, mainly on language, below:

Line 167 “employ them indistinctly” Wording does not make sense

Response: True, we changed it to “we **use** them **interchangeably**”

Line 179 “as well as” should be “or”.

Response: OK

Line 192 - 194: that result could also suggest the presence of a repressor (no need to change the text).

Yes, that could be a possibility, but the localized expression we observe would suggest that Abd-B would be repressing the repressor. As this explanation is too complex we left the text as it was.

Line 201: add “that” in observed “that” activated.

Response: OK

Line 256: “To find out *enD* requirement during development” Poor wording

“induced” should be “generated”

Response: We changed this sentence to: “To find out **what is *enD* required for** during development, we **generated *enD*Δ**, a deletion **in the endogenous gene** encompassing the *enD0.4* element that eliminates En expression from the A8a compartment”

I would also add “in the endogenous gene” to distinguish that from a reporter transgene mutation, which is really different.

Response: OK

Line 280: add melanogaster.

Response: OK

Line 315: remove “it” from “As it could”.

Response: OK

Line 317: Start new sentence with “This may be”

Response: OK

Line 318: remove “it” before as.

Response: OK

Line 326: change “sperms” to “sperm”

Response: OK

Line 358 – 360: I do not know of Abd-B ever being implicated in epigenetic regulation. Is this a pure speculation or is it based on a known activity of Abd-B?

Response: We now say that this is speculation

Line 438: remove “it” after as.

Response: OK

Reviewer #3 (Remarks to the Author):

The manuscript is greatly improved in response to reviewer feedback. **Readability of the abstract is still a major issue** and I would encourage the authors to think more carefully about the following:

1. Lines 32 – 34 of abstract

“We show that a *Drosophila* expression novelty, the activation of an engrailed cis-regulatory element for sperm liberation, has influenced its organ of origin due to gene network sharing. This network “interlocking” resulted in the posterior compartment determining factor Engrailed expressed in the anterior compartment of the A8 segment without any functional consequence.”

Are not very clear to understand for someone reading the abstract. Introducing the term “interlocking” in the abstract in quotes without defining it will disinclude less-informed readers. The other problem is that the expression novelty is not defined – are you defining the activation of en in the anterior compartment of the spiracle an expression novelty? Or is it the function that is required for sperm liberation? Clearly it is both, but the abstract does not specify with any clarity, while instead trying to drive high level concepts the reader likely won’t understand. I would also strongly consider a more explicit statement on what GRN co-option is, as the first sentence of the abstract requires an understanding that most readers will not be familiar with. It is necessary to spend two sentences setting up the contrast between single gene co-options and whole network co-options? In this reviewers mind, I find the dichotomy unhelpful if none of the terms being discussed are defined sufficiently in the abstract. I’ll note that the summary presented at the end of the introduction (lines 102-118) is much more readable, and could be used as a guide to condense into a more readable abstract.

Response: We have modified the abstract to make it more readable. As a result, its length has increased to 185 words:

New abstract:

The re-use of genes in new organs forms the base of many evolutionary novelties. A well-characterized case is the recruitment of the posterior spiracle gene-network to the *Drosophila* male genitalia. Here we find that this network has also been co-opted to the testis mesoderm where is required for sperm liberation, providing an example of sequentially repeated developmental co-options. Associated to this co-option event, an evolutionary expression novelty appeared, the activation of the posterior segment determinant Engrailed to the anterior A8 segment controlled by shared testis and spiracle regulatory elements. Enhancer deletion shows that A8 anterior Engrailed activation is not required for spiracle development but only required in the testis. Our study presents an example of pre-adaptive developmental novelty: the activation of the Engrailed transcription factor in the anterior compartment of the A8 segment where, despite having no specific function, opens the possibility of this developmental factor acquiring one. We propose that recently co-opted networks become interlocked, so that any change to the network because of its function in one organ, will be mirrored by other organs even if it provides no selective advantage to them.

2. Ln 61 typo – ananassae.

Response: Corrected

3. Ln 326-328 I still think it unlikely that expression of engrailed itself caused differences in testis/sperm morphology. Is the morphology of the enD deletion testis more like that of *Episyrphus balteatus*? If so, it is not noted in the paper. It strikes me as a fanciful point to end the results section on.

Response: We are not claiming that engrailed expression in the testis is responsible for the change in morphology, there must be many factors involved in this. To clarify this issue, we have modified the sentence that now reads:

“In fact, *Drosophila* and *Episyrphus* testis morphology is very different and even the sperm of *Episyrphus balteatus* have shorter tails as shown by staining the axoneme with Axo49 antibody (Fig. 6b,d,f), suggesting that, among other factors, the posterior spiracle gene network co-option may have contributed to their morphological divergence.”

4. The final paragraph of the discussion (lines 397-440) is very long! I would recommend splitting the two ideas it contains. Maybe break to a new paragraph at line 414, or maybe line 425.

Response: As recommended, we have split the sentence at line 425.